# Generalizing Multi-Scale Time-Series Modeling with a Single Operator

**Cheonwoo Lee** [1]  **Dooho Lee** [1]  **Doyun Choi** [2]  **Jaemin Yoo** [2]

## Abstract

Multi-scale modeling has emerged as an effective design principle for time-series forecasting by capturing temporal dynamics at multiple resolutions. As no principled foundation has been established in the literature, we unify existing scaling methods into a *scaling operator family*, revealing a fundamental limitation of existing approaches: reliance on fixed and discrete scaling. To address this limitation, we propose SIGMA (Single Generalized Multi-scale Architecture), which enables distance-aware scaling via the learnable discrete Gaussian (LDG) kernel grounded in scale-space theory. We evaluate SIGMA comprehensively on long- and short-term forecasting benchmarks against state-of-the-art multi-scale baselines. SIGMA outperforms all competitors on both tasks, achieving the best performance in 13 out of 16 long-term evaluation settings. Beyond accuracy, SIGMA significantly improves training speed by up to 5.3 times and reduces memory consumption by up to 3.8 times over the strongest competitors. Code is available at https://github.com/cheonwoolee/SiGMA.

## 1. Introduction

Time-series forecasting is widely applied in diverse domains, including energy systems (Deb et al., 2017), weather prediction (Dimri et al., 2020), and traffic management (Tedjopurnomo et al., 2020). Despite its importance, accurate forecasting is challenging due to the complex temporal structures of real-world time series (Kolambe & Arora, 2024). Inspired by the success of deep learning, recent work has introduced neural forecasting methods, including MLP-based methods (Ekambaram et al., 2023; Zeng et al., 2023; Yi et al., 2023; Challu et al., 2023) and Transformer-based

methods (Nie et al., 2023; Liu et al., 2024; Shi et al., 2025).

Within this scope, *multi-scale modeling* has emerged as a powerful design principle. By constructing temporal dynamics at multiple resolutions, it explicitly models cross-scale interactions to enhance predictive performance. Specifically, coarse-grained scaling captures long-term dynamics, whereas fine-grained scaling preserves high-frequency fluctuations, yielding a more comprehensive representation (Chen et al., 2023). Building upon this principle, recent work has introduced diverse strategies from simple multi-rate sampling to hierarchical and frequency-aware mechanisms (Zhao et al., 2024; Wang et al., 2024; 2025; Naghashi et al., 2025; Hu et al., 2025a; Yang et al., 2025).

We visualize popular scaling operators in Figure 1. Each scaling operator takes an integer scale parameter $s$ to transform the given sequence and applies the same scale uniformly across all timesteps within a single transformation. This reveals a key structural limitation: scale is discretized and shared across all timesteps. In practice, characteristic time scales, such as dominant periods or decay rates, are real-valued and may vary over time. As a result, this design restricts the ability to represent temporal dynamics that vary smoothly across resolutions and introduces implicit boundaries between representations at different scales. Since most existing approaches rely on these scaling operators (Wang et al., 2023; 2024; Hu et al., 2025a), such limitations are inherent to their multi-scale constructions.

To make these limitations explicit and to address them systematically, we first define a *family of scaling operators* that unifies the scaling methods studied in the literature (see Table 1). Within this framework, we define two fundamental requirements, *non-expansiveness* and *energy reduction*, that any valid scaling operator must satisfy. We further show that trivial operators on sequences, such as constant mappings or permutations, fail to meet these criteria.

Building on this unified view, we propose SIGMA (Single Generalized Multi-scale Architecture), a principled framework for multi-scale modeling. Specifically, we introduce a novel concept of a *generalized scaling operator family*, which generalizes existing scaling operators by allowing continuous scale parameters while preserving their structural properties. SIGMA adopts a learnable discrete Gaussian (LDG) kernel grounded in scale-space theory as an

[1]School of Electrical Engineering, KAIST, Daejeon, Republic of Korea [2]Department of Computer Science and Engineering, Seoul National University, Seoul, Republic of Korea. Correspondence to: Jaemin Yoo <jaeminyoo@snu.ac.kr>.

*Proceedings of the $43^{rd}$ International Conference on Machine Learning*, Seoul, South Korea. PMLR 306, 2026. Copyright 2026 by the author(s).

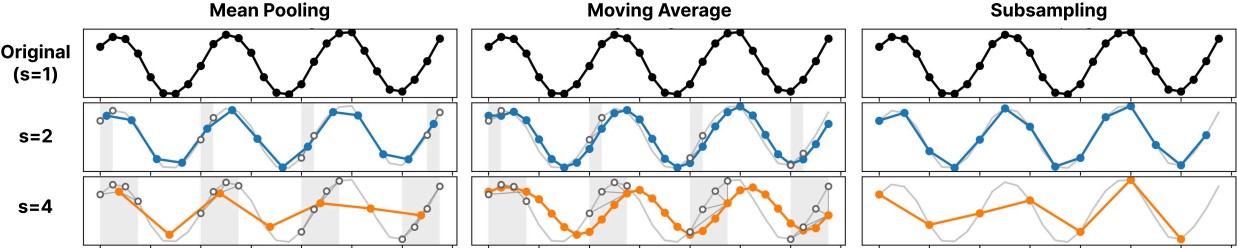

*Figure 1.* Examples of popular scaling operators used in existing methods. Each operator applies a discrete scaling parameter $s$ uniformly across all timesteps to transform the input into coarser representations, often creating an abstraction that is mismatched with the dominant periods or decay rates of the time series. Our goal is to design a learnable, dynamic scaling through a generalized framework.

instance of such a generalized scaling operator family. As a result, it allows for multiple, continuous, and learnable scale parameters within a single operator, preserving a coherent cross-scale structure directly at the representation level.

We evaluate SIGMA extensively on long- and short-term forecasting benchmarks and observe consistent improvements over state-of-the-art multi-scale models. In particular, on long-term tasks, SIGMA achieves the best performance in 13 out of 16 evaluation settings with up to 7.8% lower error than the strongest baselines. Beyond accuracy, SIGMA delivers clear efficiency gains, reducing memory usage by up to $3.8\times$ and increasing training speed by $5.3\times$. We further support these findings with comprehensive ablation and case studies, confirming that SIGMA provides a principled and efficient framework for multi-scale time-series modeling.

Our contributions are summarized as follows:

- **Unified Framework:** We introduce a novel concept of a scaling operator family, which provides a principled foundation for multi-scale time-series modeling and clarifies the structural limitations of existing approaches.

- **Generalized Architecture:** We propose SIGMA, a principled architecture for multi-scale modeling. It enables distance-aware scaling through the learnable discrete Gaussian kernel with dynamic scaling parameters.

- **Empirical Validation:** We show that SIGMA achieves state-of-the-art performance on extensive long- and short-term forecasting benchmarks while substantially reducing computational complexity compared to prior multi-scale models. We further conduct comprehensive ablation and case studies to provide deeper insights on SIGMA.

## 2. Problem Definition and Related Work

**Problem Definition**  Given a time series, let $\boldsymbol{x} \in \mathbb{R}^L$ denote an input sequence of length $L$, and let $\boldsymbol{y} \in \mathbb{R}^T$ denote the corresponding future trajectory of length $T$. We denote the input space as $\mathcal{X} \subset \mathbb{R}^L$ and the target space as $\mathcal{Y} \subset \mathbb{R}^T$. Our goal is to learn a forecasting model $h : \mathcal{X} \to \mathcal{Y}$ that maps a past window $\boldsymbol{x}$ to a $T$-step forecast of its future $\boldsymbol{y}$.

**Assumption on Data**  Throughout this work, we focus on non-trivial datasets whose forecastability $\phi$ is strictly less than 1. The forecastability of a time series is defined as one minus the normalized spectral entropy of its Fourier decomposition (Goerg, 2013), formally given as

$$\phi(\boldsymbol{x}) = 1 - \frac{H(\boldsymbol{x})}{\log(2\pi)} \in [0, 1], \tag{1}$$

$$H(\boldsymbol{x}) = - \int_{-\pi}^{\pi} p_{\boldsymbol{x}}(\omega) \log p_{\boldsymbol{x}}(\omega) \, d\omega, \tag{2}$$

where $p_{\boldsymbol{x}}(\omega)$ denotes the normalized spectral density of $\boldsymbol{x}$ on $[-\pi, \pi]$, characterizing how concentrated the spectrum of $\boldsymbol{x}$ is. Most real-world datasets, including every benchmark used in our experiments, satisfy $\phi < 1$ (see Appendix E). This serves as a mild theoretical assumption and does not restrict the practical applicability of our framework. In general, larger forecastability $\phi$ indicates a more concentrated spectrum and thus a stronger predictable structure (Wang et al., 2024; 2025), which is desirable for forecasting.

**Time Series Forecasting**  The field of time series forecasting has seen a significant evolution from traditional statistical methods to modern deep learning architectures. While statistical models like ARIMA (Box et al., 2015) and state-space models (Durbin & Koopman, 2012) are still used, they are often limited in their ability to capture complex nonlinear and long-range dependencies (Zhou et al., 2021). Among deep forecasting models that learn representations directly from raw sequences, multi-layer perceptron architectures have recently been recognized for their effectiveness and efficiency (Ekambaram et al., 2023; Zeng et al., 2023; Yi et al., 2023; Challu et al., 2023). Concurrently, Transformer-based approaches have demonstrated strong empirical results, particularly for long-horizon forecasting tasks (Nie et al., 2023; Liu et al., 2024; Shi et al., 2025).

**Multi-Scale Design in Time Series**  Time series often exhibit complex patterns across diverse temporal scales, from rapid fluctuations to long-term trends. To capture such heterogeneity, recent work has incorporated explicit multi-scale mechanisms to disentangle and integrate temporal dependencies. These methods include hierarchical decompositions

*Table 1.* Representative scaling operator families with associated scale parameters $s$. Each family simplifies the sequence in a structured manner as the scale parameter $s$ increases, where $k \in [1, L_s]$ is the element index and $L_s$ denotes the length of the output sequence. For wavelet decomposition, $f(\boldsymbol{x}|s)$ yields the approximation coefficients at level $s$ using the scaling function $\phi$.

| Operator $f$ | Equation | Scale parameter $s$ | $L_s$ | Usage |
|---|---|---|---|---|
| Average pooling | $[f(\boldsymbol{x}|s)]_k = (1/s) \sum_{i=0}^{s-1} x_{ks+i}$ | Window length | $P/s$ | Wang et al. (2024) |
| Max pooling | $[f(\boldsymbol{x}|s)]_k = \max_{0 \le i < s} x_{ks+i}$ | Window length | $P/s$ | Challu et al. (2023) |
| Moving average | $[f(\boldsymbol{x}|s)]_k = (1/s) \sum_{i=0}^{s-1} x_{k-i}$ | Window length | $P$ | Wang et al. (2023) |
| Subsampling | $[f(\boldsymbol{x}|s)]_k = x_{ks}$ | Stride | $P/s$ | Liu et al. (2022a) |
| Segmentation | $[f(\boldsymbol{x}|s)]_k = x_k$ | Length divisor | $P/s$ | Wu et al. (2023) |
| Wavelet decomposition | $[f(\boldsymbol{x}|s)]_k = \langle \boldsymbol{x}, \phi_{s,k} \rangle$ | Decomposition level | $P/2^s$ | Murad et al. (2025) |

that construct multi-resolution representations via downsampling (Liu et al., 2022b; Challu et al., 2023), frequency-domain or wavelet-based analyses that emphasize periodic structure through spectral transforms (Wu et al., 2023; Cai et al., 2024; Murad et al., 2025), and aggregation schemes that fuse representations across scales (Wang et al., 2023; 2024; 2025). However, existing multi-scale methods rely on fixed and discrete scaling strategies to generate multi-scale representations, which limits adaptability to diverse and continuously varying temporal patterns. This highlights the need for more flexible and principled approaches to multi-scale modeling.

## 3. Foundations of Scaling in Time Series

In time series analysis, *scaling* denotes transformations that change the temporal resolution of time series. As illustrated in Figure 1, existing methods employ common operations such as downsampling and moving averages with discrete scale parameters to produce representations at different resolutions. However, these operations are typically introduced heuristically, offering limited insight into why particular scales are appropriate or how they relate to one another.

To address this gap, we introduce a rigorous definition of *scaling operator families*, providing a unified mathematical framework for analyzing scaling operations in time series. To the best of our knowledge, this is the first work to establish such a foundation. All proofs are given in Appendix A.

**Definition 3.1** (Scaling operator family). A set of transformations $\mathcal{F} = \{f(\boldsymbol{x}|s) \mid s \in \mathbb{Z}_+\}$ such that $f(\cdot|s) : \mathcal{X} \to \mathcal{X}_s \subset \mathbb{R}^{L_s}$, conditioned on a positive integer $s \in \mathbb{Z}_+$, is a scaling operator family if it satisfies

- *Non-expansiveness*: For $s_i \in \mathbb{Z}_+$ and $\boldsymbol{x}, \boldsymbol{x}' \in \mathcal{X}$, $\|f(\boldsymbol{x}|s_i) - f(\boldsymbol{x}'|s_i)\|_2 \le \|\boldsymbol{x} - \boldsymbol{x}'\|_2$.
- *Energy reduction*: For any $s_i, s_j \in \mathbb{Z}_+$ such that $s_i = m \cdot s_j$ for some $m > 1$, $\|f(\boldsymbol{x}|s_i)\|_2 \le \|f(\boldsymbol{x}|s_j)\|_2$ for all $\boldsymbol{x} \in \mathcal{X}$ and $\|f(\boldsymbol{x}'|s_i)\|_2 < \|f(\boldsymbol{x}'|s_j)\|_2$ for some $\boldsymbol{x}' \in \mathcal{X}$.

The scale parameter $s$ determines the level of scaling and also the output dimension $L_s$.

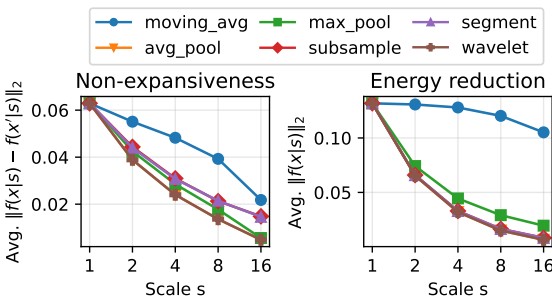

*Figure 2.* The non-expansiveness and energy reduction of the six scaling operator families on the Traffic dataset. All these operators satisfy the two essential properties stated in Definition 3.1.

**Theorem 3.2.** *Popular sequence operations used in recent work (Liu et al., 2022a; Challu et al., 2023; Wu et al., 2023; Wang et al., 2023; 2024; Murad et al., 2025), such as max-or mean-pooling (Zheng et al., 2014), subsampling (Hannan, 2009), moving averages (Box et al., 2015), segmentation (Keogh et al., 2004) and wavelet decompositions (Percival & Walden, 2000), are scaling operator families.*

Definition 3.1 formalizes the expected behavior of scaling. Scaling operators are *non-expansive*, as they are designed to capture diverse properties from a sequence without adding external information. At the same time, the *energy reduction* property formalizes the intuition that time series with coarser scales should be simpler than finer ones along multiplicative chains of scale parameters: as the scale parameter $s$ increases, the result exhibits less overall energy, with small fluctuations being smoothed out.

Table 1 shows the operator families stated in Theorem 3.2 with their associated scale parameters. These operators can be grouped into two categories based on how they transform the given sequence: *(i) smoothing operators*, which attenuate fine-grained variability by summarizing adjacent observations (e.g., pooling and moving average), and *(ii) structural operators*, which progressively decompose or resample the signals by enforcing a specific structure (e.g., subsampling and wavelet decomposition). Previous works have typically used the term scaling *only for one group*, but our Definition 3.1 suggests that both categories of operators

can be understood through the same lens in terms of the non-expansiveness and energy reduction properties.

To empirically verify that the operator families in Table 1 satisfy both properties stated in Definition 3.1, we evaluate their average pairwise contraction and induced average energy on the Traffic dataset. As shown in Figure 2, all operator families exhibit strictly smaller output differences than input differences, confirming non-expansiveness, and their energies decrease monotonically over multiplicative scales, confirming energy reduction. Additional experiments on other datasets are provided in Appendix B.

**Theorem 3.3.** *Trivial sequence operations (Horn & Johnson, 2012) such as constant mappings, permutations, additive shifts, scalar multiplications, and general linear transformations do not form scaling operator families.*

Theorem 3.3 highlights that not all parameterized transformations qualify as scaling operator families. For example, while a scalar multiplication can change the scale of each observation, it does not guarantee energy reduction and thus falls outside our definition. Putting Theorem 3.2 and Theorem 3.3 together, we claim that our definition of scaling operator families is designed carefully to include only the ones that have been considered "scaling" in the literature, providing more interpretable and theoretically grounded approaches to multi-scale time series modeling.

# 4. SIGMA: Generalized Multi-Scale Modeling

We introduce SIGMA, a principled approach to multi-scale time series modeling. We first generalize the family of scaling operators to support multiple continuous scale parameters (Sec. 4.1). Then, we introduce the learnable discrete Gaussian (LDG) kernel grounded in the scale-space theory as an effective instantiation of the generalized scaling operator (Sec. 4.2). Lastly, we introduce our implementation design based on these theoretical foundations (Sec. 4.3). Detailed proofs for all theorems are provided in Appendix A.

## 4.1. Generalized Scaling Operator Family

The unified foundation for scaling operator families reveals their essential restriction: they are parameterized by a single, discrete scale parameter $s \in \mathbb{Z}_+$ that is fixed a priori and applied uniformly across the sequence. Such a restriction often makes scaling operators misaligned with real-world data, where characteristic time scales vary across datasets and may evolve within a single sequence. To address this limitation, we generalize Definition 3.1 to allow multiple continuous scale parameters $s \in \mathbb{R}_+^M$ for each $f$.

**Definition 4.1** (Generalized scaling operator family)**.** A set of transformations $\mathcal{F} = \{f(\boldsymbol{x}|\boldsymbol{s}) \mid \boldsymbol{s} \in \mathbb{R}_+^M\}$ such that $f(\cdot|\boldsymbol{s}) : \mathcal{X} \to \mathcal{X}_{\boldsymbol{s}} \subset \mathbb{R}^{L_{\boldsymbol{s}}}$ is a generalized scaling operator family if it satisfies

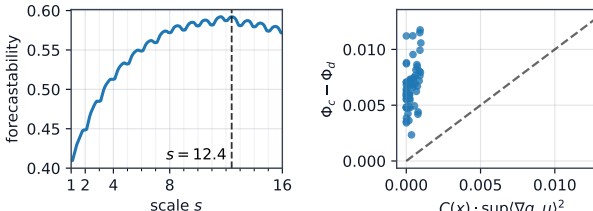

*Figure 3.* Empirical validation of Theorem 4.2 on the Traffic dataset. Forecastability is maximized at a non-discrete scale, and the expressivity gap $\Phi_c - \Phi_d$ exceeds the theoretical lower bound.

- *Consistency*: For $s \in \mathbb{Z}_+$, $\{f(\boldsymbol{x}|s\boldsymbol{1})\}$ is a scaling operator family.
- *Differentiability*: For any $\boldsymbol{x} \in \mathcal{X}$, $f(\boldsymbol{x}|\boldsymbol{s})$ is continuously differentiable with respect to $\boldsymbol{s}$.

The output dimension $L_{\boldsymbol{s}}$ depends on the set of scale parameters $\boldsymbol{s} = (s_1, \cdots, s_M)$.

The consistency condition ensures that Definition 4.1 strictly generalizes discrete scaling operator families. The differentiability condition guarantees that changes in scale parameters induce smooth changes in the transformed signal.

**Theorem 4.2.** *Let $\mathcal{S}_d \subset \mathbb{Z}^+$ be the discrete set of scale parameters and $\mathcal{S}_c \subset \mathbb{R}_+^M$ be a compact set containing $\mathcal{S}_d$. For a given scaling operator family $f$ and forecastability measure $\phi$, define $g_{\boldsymbol{x}}(\boldsymbol{s}) = \phi(f(\boldsymbol{x}|\boldsymbol{s}))$,*

$$\Phi_c(\boldsymbol{x}) = \max_{\boldsymbol{s} \in \mathcal{S}_c} g_{\boldsymbol{x}}(\boldsymbol{s}), \quad and \quad \Phi_d(\boldsymbol{x}) = \max_{s \in \mathcal{S}_d} g_{\boldsymbol{x}}(s\boldsymbol{1}).$$

*Let $\Pi_{\boldsymbol{1}} = \{\boldsymbol{u} \in \mathbb{R}^M : \boldsymbol{u}^\top \boldsymbol{1} = 0, \|\boldsymbol{u}\|_2 = 1\}$. Then,*

$$\Phi_c(\boldsymbol{x}) - \Phi_d(\boldsymbol{x}) \geq C(\boldsymbol{x}) \sup_{\boldsymbol{u} \in \Pi_{\boldsymbol{1}}} \langle \nabla_{\boldsymbol{s}} g_{\boldsymbol{x}}(\boldsymbol{s})|_{\boldsymbol{s}=t\boldsymbol{1}}, \boldsymbol{u} \rangle^2,$$

*for all $\boldsymbol{x} \in \mathcal{X}$, with some constant $C(\boldsymbol{x}) > 0$ and $t \in \mathbb{R}$.*

Theorem 4.2 formalizes a quantization gap in multi-scale forecasting. The set $\Pi_{\boldsymbol{1}}$ consists of directions that lie outside the representational capacity of a discrete scaling operator family. If the forecastability function $g$ increases along any direction in $\Pi_{\boldsymbol{1}}$, then this improvement cannot be captured by any $s \in \mathcal{S}_d$. In such cases, a generalized scaling operator family yields strictly larger forecastability. Intuitively, restricting scale parameters to discrete values introduces an irreducible mismatch between the operator's effective scale and the signal's intrinsic temporal structure; discrete scaling operator families incur an irreducible loss in forecastability relative to generalized families.

To empirically validate Theorem 4.2, we evaluate the average forecastability and quantization gap on the Traffic dataset. As shown in Figure 3, the left panel indicates that forecastability is maximized at a non-discrete scale, implying that the maximum forecastability over the continuous scale set, $\Phi_c$, exceeds that over the discrete scale set, $\Phi_d$.

Furthermore, the right panel shows that the quantization gap, $\Phi_c - \Phi_d$, consistently exceeds the theoretical lower bound across all samples $\boldsymbol{x}$, thereby providing empirical support for Theorem 4.2. Additional results on the other datasets are provided in Appendix C.

As Definition 4.1 is not intuitive at first glance, it may be nontrivial to design an operator that satisfies both properties. We thus provide a guideline for designing generalized scaling operator families in Appendix D (e.g., generalized mean-pooling). The key idea is to obtain generalized operators through a smooth expansion of discrete scaling operators with respect to their scale parameters. Concretely, this is realized by interpolating discrete operators through smooth functions of scale, ensuring exact recovery at integer scales and preserving the differentiability condition.

### 4.2. Learnable Discrete Gaussian as a Scaling Operator

Among various constructions that can realize a generalized scaling operator family, we propose adopting the *learnable discrete Gaussian* (LDG) kernel $k$ (Lindeberg, 2002):

$$k(\boldsymbol{x}|\boldsymbol{s}) = \boldsymbol{K}(\boldsymbol{s})\boldsymbol{x}, \qquad [\boldsymbol{K}(\boldsymbol{s})]_{i,j} = e^{-s_d}I_d(s_d), \quad (3)$$

where $\boldsymbol{K}(\boldsymbol{s}) \in \mathbb{R}^{L \times L}$ is the kernel matrix with $L$ being the length of the sequence $\boldsymbol{x}$, $[\cdot]_{i,j}$ denotes the $(i, j)$-th element of a matrix, $I_n(\cdot)$ is the modified Bessel function of the first kind of integer order $n$, and $d = |i - j|$ denotes the temporal distance between positions $i$ and $j$.

Intuitively, the LDG kernel in Equation 3 performs a smooth, locality-aware aggregation of the input sequence. Each output element is computed as a weighted average of its neighboring elements with symmetrically decaying weights. The parameter $s_d$ controls how broadly information is aggregated at distance $d$: small values concentrate the weights near the center, preserving local variations, while larger values spread the weights over a wider temporal neighborhood, producing smoother representations.

Unlike classical discrete Gaussian filtering, we model the scale parameters $\boldsymbol{s}$ as learnable and position-dependent. As a result, $k$ applies dynamic scaling to each element of $\boldsymbol{x}$, allowing the operator to adaptively control the extent of temporal aggregation at each time step. Since the LDG kernel can apply heterogeneous smoothing across the sequence, it avoids the structural limitations of discrete designs. These scale parameters are optimized from data via gradient-based learning with respect to the downstream forecasting loss.

Figure 4 illustrates how the LDG kernel actually works on a time series; it yields smooth, scale-controlled transformations at each time step with learnable scale parameters. At the same time, it provides a symmetric and effectively

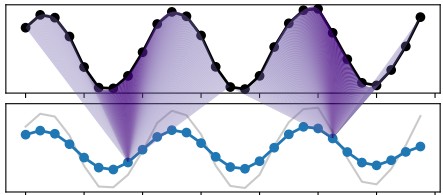

*Figure 4.* The LDG kernel performs smoothing with continuous distance-aware scales. It induces an effectively unbounded receptive field, enabling long-range dependency modeling.

unbounded receptive field for capturing long-range dependencies, which is its unique characteristic.

**Theorem 4.3.** *The LDG kernel family, $\{k(\cdot|\boldsymbol{s}) \mid \boldsymbol{s} \in \mathbb{R}_+^M\}$, is a generalized scaling operator family.*

**Theorem 4.4.** *The LDG kernel is the unique generalized scaling operator family of symmetric kernels that satisfies the discrete scale-space axioms (Lindeberg, 2002).*

Theorem 4.3 establishes that this kernel constitutes a valid generalized scaling operator family, confirming its suitability for principled scaling. Moreover, this Gaussian instantiation is the unique operator determined by the discrete scale-space axioms as shown in Theorem 4.4. This finding aligns our method with the scale-space theory, a principled framework for multi-resolution signal analysis originating in computer vision (Witkin, 1987; Lindeberg, 2013). We empirically examine alternative kernel choices in Section 5.2, showing that replacing the LDG kernel with other smoothing or learnable operators consistently degrades performance.

### 4.3. Lightweight Forecasting Module

Many existing multi-scale forecasting models integrate information across scales after applying the scaling operator, such as weighted summation or aggregation modules (Wu et al., 2023; Cai et al., 2024), or cross-scale mixing architectures that dynamically couple representations at different temporal resolutions (Hu et al., 2025a; Wang et al., 2024; 2025). While these designs improve expressivity under discrete scaling operator families, they typically incur additional parameters and computational overhead.

An essential strength of our learnable scaling operation is that additional multi-scale interaction is not required in the forecasting module, since the LDG kernel already preserves coherent cross-scale structure at the representation level. Accordingly, prediction can be performed using an arbitrary nonlinear mapping. We therefore adopt a lightweight multi-layer perceptron (MLP), which is parameter-efficient, fully parallel, and empirically sufficient for real-world data, as confirmed by the ablation results in Section 5.2.

Formally, given an input sequence $\boldsymbol{x} \in \mathbb{R}^L$, we first produce $d$-dimensional embeddings $\boldsymbol{X} = \text{Embed}(\boldsymbol{x}) \in \mathbb{R}^{L \times d}$ as

*Table 2.* Long-term forecasting results across eight datasets with horizons $T \in \{96, 192, 336, 720\}$ and input length fixed at 96. Results are averaged across all horizons, while the full results are provided in Appendix F. SIGMA achieves the smallest forecasting errors in 13 out of 16 evaluation settings and the second-best in 2 cases.

| Models | SIGMA (Ours) | | AMD (2025a) | | MultiPatch. (2025) | | WPMixer (2025) | | TimeMixer (2024) | | MSGNet (2024) | | MICN (2023) | | TimesNet (2023) | | Pyra. (2022b) | |
|---|---|---|---|---|---|---|---|---|---|---|---|---|---|---|---|---|---|---|
| | MSE | MAE | MSE | MAE | MSE | MAE | MSE | MAE | MSE | MAE | MSE | MAE | MSE | MAE | MSE | MAE | MSE | MAE |
| Weather | 0.247 | **0.273** | 0.263 | 0.286 | 0.255 | 0.278 | **0.245** | **0.273** | 0.246 | 0.276 | 0.258 | 0.283 | 0.266 | 0.315 | 0.262 | 0.288 | 0.285 | 0.347 |
| Electricity | **0.175** | **0.269** | 0.208 | 0.289 | 0.198 | 0.282 | 0.194 | 0.282 | 0.185 | 0.274 | 0.199 | 0.307 | 0.187 | 0.298 | 0.194 | 0.294 | 0.298 | 0.390 |
| Traffic | **0.458** | **0.302** | 0.546 | 0.344 | 0.497 | 0.329 | 0.527 | 0.343 | 0.501 | 0.318 | 0.654 | 0.384 | 0.543 | 0.320 | 0.627 | 0.334 | 0.685 | 0.384 |
| Exchange | 0.353 | **0.400** | 0.358 | 0.401 | 0.387 | 0.419 | 0.376 | 0.408 | 0.384 | 0.414 | 0.431 | 0.446 | **0.345** | 0.424 | 0.416 | 0.443 | 1.181 | 0.854 |
| ETTh1 | **0.443** | **0.433** | 0.447 | 0.434 | **0.443** | 0.440 | 0.451 | 0.442 | 0.455 | 0.444 | 0.459 | 0.458 | 0.572 | 0.531 | 0.480 | 0.468 | 0.879 | 0.731 |
| ETTh2 | **0.376** | 0.402 | **0.376** | **0.400** | 0.379 | 0.406 | 0.390 | 0.410 | 0.389 | 0.410 | 0.402 | 0.421 | 0.582 | 0.527 | 0.410 | 0.424 | 4.086 | 1.621 |
| ETTm1 | **0.383** | **0.397** | 0.395 | 0.399 | 0.385 | 0.400 | 0.385 | **0.397** | 0.385 | 0.398 | 0.401 | 0.412 | 0.399 | 0.427 | 0.410 | 0.415 | 0.737 | 0.614 |
| ETTm2 | **0.276** | **0.322** | 0.285 | 0.328 | 0.283 | 0.328 | 0.279 | 0.324 | 0.281 | 0.327 | 0.289 | 0.330 | 0.354 | 0.397 | 0.298 | 0.333 | 1.462 | 0.846 |

done in (Wu et al., 2021; Wang et al., 2024; 2025). Based on these embeddings, our multi-scale forecasting model is succinctly represented by the following equation:

$$\hat{y} = \boldsymbol{W}_1(\text{MLP}(\boldsymbol{H}) + \boldsymbol{H})\boldsymbol{W}_2, \qquad (4)$$

$$\boldsymbol{H} = \boldsymbol{K}(\boldsymbol{s})\boldsymbol{X} \parallel (\boldsymbol{I} - \boldsymbol{K}(\boldsymbol{s}))\boldsymbol{X} \in \mathbb{R}^{2L \times d}, \qquad (5)$$

where $\boldsymbol{W}_1 \in \mathbb{R}^{T \times 2L}$ and $\boldsymbol{W}_2 \in \mathbb{R}^{d \times 1}$ are projection heads and $\boldsymbol{s} \in \mathbb{R}_+^L$ denotes learnable scale parameters. These scale parameters are optimized as dataset-level parameters during training and kept fixed at inference.

One notable design decision is that we decompose the input into a smoothed component $\boldsymbol{K}(\boldsymbol{s})\boldsymbol{X}$ and a residual component $(\boldsymbol{I} - \boldsymbol{K}(\boldsymbol{s}))\boldsymbol{X}$ analogous to the classical trend-seasonal decomposition (Cleveland et al., 1990). To further stabilize optimization and preserve scale-specific information, we incorporate a skip connection that adds each scale's input back to its transformed output (He et al., 2016). We also adopt channel independence, processing each variable separately, which makes the architecture naturally applicable to multivariate settings (Zeng et al., 2023). We further apply reversible instance normalization to each time series variable to remove distribution shifts (Kim et al., 2022).

## 5. Experiments

We conduct a comprehensive empirical evaluation of SIGMA on standard long-term and short-term forecasting datasets against eight state-of-the-art baseline models. We also provide deeper analyses to investigate its efficiency, robustness, and underlying design principles.

**Datasets** We evaluate SIGMA on both long-term and short-term forecasting tasks, following the TSLib benchmark (Wang et al., 2026). The evaluated datasets include (long-term) Weather, ETT (ETTh1, ETTh2, ETTm1, ETTm2), Electricity, Exchange, Traffic, and (short-term) M4. Following the standard protocol (Zhou et al., 2021),

we partition all datasets into training, validation, and test sets in chronological order, using a 6:2:2 ratio for the ETT datasets and a 7:1:2 ratio for the others. Consistent with previous work, the observation window is fixed to $L = 96$, and forecasting horizons are set to $T \in \{96, 192, 336, 720\}$ for long-term forecasting tasks (Wu et al., 2023). As the M4 benchmark consists of 100,000 univariate time series collected at multiple temporal frequencies ranging from yearly to hourly, we follow the official benchmark protocol, where the prediction lengths are fixed according to the frequency of each series (Makridakis et al., 2018). Comprehensive details for each dataset are provided in Appendix E.

**Baselines** We compare SIGMA with eight state-of-the-art baselines for time series forecasting based on multi-scale modeling strategies: AMD (Hu et al., 2025a), MultiPatch-Former (Naghashi et al., 2025), WPMixer (Murad et al., 2025), TimeMixer (Wang et al., 2024), MSGNet (Cai et al., 2024), MICN (Wang et al., 2023), TimesNet (Wu et al., 2023), and Pyraformer (Liu et al., 2022b).

**Experimental Settings** We adopt evaluation metrics used in previous work for each benchmark. For long-term forecasting, we use mean squared error (MSE) and mean absolute error (MAE) (Wu et al., 2023). For short-term forecasting, we report symmetric mean absolute percentage error (SMAPE), mean absolute scaled error (MASE), and the overall weighted average (OWA) (Oreshkin et al., 2020). We apply the same batch size, number of training epochs, and early-stopping strategy across all methods (Wu et al., 2023). Each experiment is repeated three times, and the averaged results are reported.

### 5.1. Comparison with Baseline Models

**Long-Term Forecasting** Table 2 demonstrates that SIGMA achieves the best performance in 13 out of 16 evaluation settings for long-term forecasting across all datasets and horizons. The improvements are particularly evident in the

*Table 3.* Short-term forecasting results in the M4 benchmark dataset with various temporal granularities. SIGMA performs the best in 11 of the 15 cases. This highlights its effectiveness in capturing multi-scale patterns across a diverse range of time series types.

| | Metric | SIGMA (Ours) | AMD (2025a) | MultiPatch. (2025) | WPMixer (2025) | TimeMixer (2024) | MSGNet (2024) | MICN (2023) | TimesNet (2023) | Pyra. (2022b) |
|---|---|---|---|---|---|---|---|---|---|---|
| Yearly | SMAPE | 13.314 | 13.447 | **13.296** | 13.632 | 13.326 | 13.354 | 14.580 | 13.482 | 14.987 |
| | MASE | **2.989** | 3.022 | 3.009 | 3.075 | 3.002 | **2.989** | 3.382 | 3.056 | 3.361 |
| | OWA | **0.783** | 0.792 | 0.785 | 0.804 | 0.785 | 0.784 | 0.871 | 0.797 | 0.881 |
| Quarterly | SMAPE | **10.060** | 10.259 | 10.166 | 10.299 | 10.281 | 10.446 | 11.389 | 10.116 | 11.706 |
| | MASE | **1.177** | 1.211 | 1.178 | 1.217 | 1.206 | 1.248 | 1.380 | 1.186 | 1.397 |
| | OWA | **0.886** | 0.907 | 0.892 | 0.911 | 0.907 | 0.929 | 1.020 | 0.892 | 1.041 |
| Monthly | SMAPE | **12.750** | 12.898 | 12.810 | 12.945 | 12.984 | 12.970 | 13.797 | 12.775 | 14.444 |
| | MASE | **0.936** | 0.952 | 0.942 | 0.959 | 0.964 | 0.976 | 1.077 | 0.945 | 1.142 |
| | OWA | **0.882** | 0.895 | 0.887 | 0.900 | 0.903 | 0.908 | 0.985 | 0.887 | 1.038 |
| Others | SMAPE | 4.867 | 4.822 | 4.849 | 4.925 | **4.739** | 5.521 | 6.123 | 4.978 | 6.115 |
| | MASE | 3.316 | 3.245 | 3.271 | 3.259 | **3.234** | 3.829 | 4.196 | 3.265 | 4.156 |
| | OWA | 1.037 | 1.021 | 1.028 | 1.028 | **1.014** | 1.174 | 1.300 | 1.048 | 1.282 |
| Weighted Average | SMAPE | **11.840** | 11.987 | 11.889 | 12.067 | 12.002 | 12.080 | 13.015 | 11.910 | 13.495 |
| | MASE | **1.585** | 1.605 | 1.591 | 1.623 | 1.604 | 1.647 | 1.836 | 1.604 | 1.864 |
| | OWA | **0.868** | 0.881 | 0.872 | 0.887 | 0.882 | 0.898 | 0.983 | 0.876 | 1.015 |

high-dimensional Electricity and Traffic datasets. SIGMA achieves a 5.4% and 7.8% reduction in MSE relative to the second-best model, respectively. On the other hand, SIGMA is less significant on the Weather dataset, which has relatively low dimensionality and high intrinsic forecastability. In such a setting, discrete scaling operators can also capture the dominant temporal structure, allowing existing methods to perform competitively. Overall, these results highlight the importance of learning distance-aware scale parameters from data to capture fine-grained temporal dynamics through principled multi-scale modeling when applied to real-world multivariate time series, with particularly strong benefits in complex, high-dimensional settings.

**Short-Term Forecasting** Table 3 shows that SIGMA also establishes clear superiority over the competitors on the M4 benchmark for short-term forecasting, achieving the best results in 11 out of 15 cases. It achieves the best average results, particularly in the Quarterly and Monthly datasets, which contain the largest number of series. On the other hand, SIGMA exhibits limited performance in the "Others" category, which lacks sufficient data, as it accounts for less than 5% of the benchmark. This suggests that learning distance-aware scale parameters is most effective when trained on sufficiently large datasets, while less-structured series favor stronger inductive biases to extract meaningful patterns. Nevertheless, SIGMA outperforms all other baselines by a large margin overall.

**Efficiency Analysis** The performance gains achieved by SIGMA are realized through a lightweight and parameter-efficient architecture. We assess efficiency in terms of per-iteration training time and memory footprint on the ETTh1 dataset under the predict-720 setting, with all methods executed on a single GPU using the same batch size. As shown

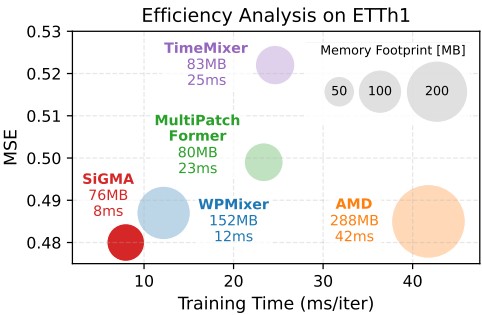

*Figure 5.* Efficiency analysis on ETTh1 with predict-720 setting. SIGMA achieves the best trade-off between accuracy and efficiency, attaining the lowest MSE while requiring substantially less training time and memory than competing multi-scale methods.

in Figure 5, SIGMA attains the fastest training time and the most compact memory footprint while achieving the lowest MSE. Specifically, SIGMA reduces memory consumption by 3.8 times and improves training speed by 5.3 times compared to AMD, the previous state-of-the-art baseline.

This significant efficiency gap comes from fundamental architectural differences. AMD stacks three downsampling stages and applies per-variable multi-scale mixing in a sequential manner. While this approach may improve accuracy over other baselines, it imposes substantial memory and runtime overhead. In contrast, SIGMA employs a single-kernel formulation that preserves full parallelism in its representation updates. Our principled simplification of multi-scale modeling yields a more favorable speed-memory trade-off without compromising modeling capacity, leading to superior overall performance. We extend this efficiency analysis to additional datasets in Appendix G, showing that SIGMA preserves faster training time with competitive memory usage on complex datasets.

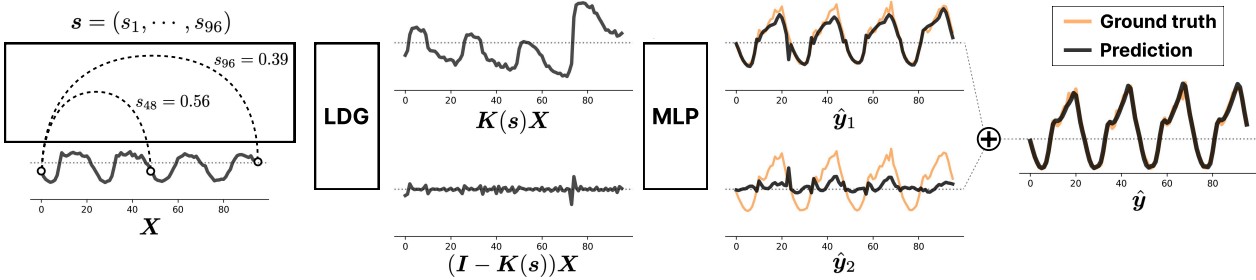

*Figure 6.* Case study on Traffic for predict-96 setting. SIGMA learns distance-aware scale parameters to adaptively control temporal smoothing via the LDG kernel, while an MLP integrates the resulting multi-scale representations to capture both long-term trends and short-term variations. By integrating these complementary signals, SIGMA achieves more accurate and effective predictions.

## 5.2. Deeper Analysis on SIGMA

**Ablation Study** To validate the design principles of SIGMA, we conduct an ablation study on ETTh1 under the predict-720 setting, comparing SIGMA against five of its variants: ① replaces the MLP predictor with the trend-seasonal mixing mechanism of TimeMixer, which is a more complicated variant of an MLP; ② changes the LDG kernel to have a single scale parameter applied to all elements; ③ changes the LDG kernel to use a sample-wise scale parameter using a two-layer MLP; ④ removes the scaling mechanism entirely and relies only on the raw input; ⑤ employs moving average, a non-learnable scale operator family; ⑥ uses unnormalized convolution, a learnable kernel that does not belong to the scaling operator family.

*Table 4.* Ablation study with predict-720 setting on ETTh1.

| Method | MSE | MAE |
|---|---|---|
| SIGMA | **0.480** | 0.468 |
| ① | 0.486 | **0.467** |
| ② | 0.489 | 0.473 |
| ③ | 0.490 | 0.474 |
| ④ | 0.492 | 0.475 |
| ⑤ | 0.493 | 0.475 |
| ⑥ | 0.524 | 0.492 |

As shown in Table 4, most modifications result in performance degradation. This indicates that the expressivity of the LDG kernel in ① and ②, the effect of scaling in ④, and the distance-aware learnable scaling in ⑤ all play a critical role. While ① achieves performance comparable to SIGMA through a different multi-scale modeling strategy, it further validates that a simpler and more parameter-efficient MLP-based formulation is sufficient for effective forecasting. ③ shows that direct sample-wise scaling does not improve performance, suggesting that its increased flexibility may introduce higher variance and greater sensitivity to noise. The significant performance drop in ⑥ demonstrates that invalid scaling operations which do not belong to the scaling operator family can be sub-optimal, as they can distort the inherent characteristics of time series as well.

**Hyperparameter Sensitivity** We further examine the sensitivity of SIGMA to the input length $L$, a key hyperparameter in multi-scale forecasting. We evaluate forecasting per-

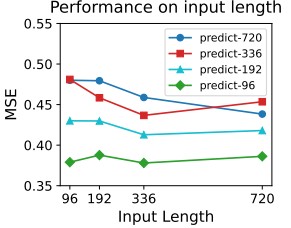
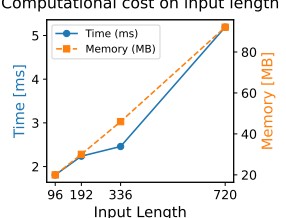

*(a)* Performance across input lengths on ETTh1.

*(b)* Computational cost across input lengths on ETTh1.

*Figure 7.* Hyperparameter sensitivity of the input length $L$ on ETTh1. (a) Larger $L$ generally benefits long-horizon forecasting, while intermediate sizes suffice for shorter horizons. (b) Training time and memory usage scale linearly with $L$.

formance (MSE) and computational cost on ETTh1 under varying lookback windows $L \in \{96, 192, 336, 720\}$.

As shown in Figure 7a, increasing the input length leads to consistent performance improvements for the long-horizon setting (predict-720), whereas for shorter horizons the best results are typically achieved with intermediate sizes. This aligns with the properties of the LDG kernel, which has an effectively unbounded receptive field, allowing longer input sequences to better capture long-range temporal structure. Figure 7b shows that memory usage grows linearly, while training time increases superlinearly with $L$. Under the current implementation, the LDG operator is applied via dense matrix multiplication, resulting in $O(L^2)$ time complexity. However, because the LDG kernel induces a Toeplitz operator, this cost could be reduced through more efficient implementations, such as truncated convolution or FFT-based multiplication. We provide further analysis in Appendix H. Overall, SIGMA achieves an effective accuracy-cost trade-off as the lookback window increases.

**Case Study** We conduct a case study on the Traffic dataset under the predict-96 setting to illustrate the behavior of SIGMA. As shown in Figure 6, SIGMA learns a scale parameter at each timestep. The LDG kernel acts as a principled smoothing operator, allowing the model to selectively em-

phasize broad trends or localized variations. The resulting representations are then transformed by a lightweight MLP, which captures richer temporal dependencies and adjusts the balance between coarse- and fine-grained components. By integrating these components, SIGMA yields predictions that closely align with the ground truth, validating the effectiveness of its multi-scale design.

## 6. Conclusion

In this work, we introduce the concept of *scaling operator families*, providing a principled foundation for understanding and unifying scaling mechanisms used in multi-scale time-series modeling. Building on this foundation, we propose SIGMA, a principled approach grounded in a *generalized scaling operator family*. SIGMA employs the learnable discrete Gaussian (LDG) kernel, which generalizes classical scaling operators by enabling continuous, distance-aware and learnable scale parameters while preserving coherent cross-scale structure at the representation level. Extensive experiments on both long- and short-term forecasting benchmarks demonstrate that SIGMA consistently outperforms state-of-the-art baselines, while substantially reducing memory consumption and computational time.

**Limitations and Future Work** A limitation of this work is that dynamic scaling parameters are learned at the dataset level, which is suboptimal when training samples are insufficient. Future work can include extending our framework to multivariate scaling and sample-specific dynamic modeling to further enhance its expressiveness and applicability. Nevertheless, our ablation results indicate that direct sample-wise parameterization requires careful design. A promising future direction is to combine shared dataset-level scales with residual sample-wise adaptations, thereby improving flexibility while preserving stability.

## Acknowledgements

This work was partly supported by the National Research Foundation of Korea (NRF) grant funded by the Korea government (MSIT) (RS-2024-00341425 and RS-2024-00406985), "Advanced GPU Utilization Support Program" funded by the Government of the Republic of Korea (Ministry of Science and ICT), and the New Faculty Startup Fund from Seoul National University.

## Impact Statement

This work contributes to the foundations of time-series forecasting through a structured multi-scale modeling approach. Its societal impact is indirect and mediated by downstream applications, and we do not foresee ethical concerns beyond those commonly associated with forecasting methods.

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

# A. Proofs

## A.1. Proof of Theorem 3.2

**Lemma A.1.** *If $T : \mathbb{R}^m \to \mathbb{R}^n$ satisfies $\|T(\boldsymbol{x}) - T(\boldsymbol{y})\|_2 \le \|\boldsymbol{x} - \boldsymbol{y}\|_2$ for all $\boldsymbol{x}, \boldsymbol{y}$, then $\|T(\boldsymbol{x})\|_2 \le \|\boldsymbol{x}\|_2$ for all $\boldsymbol{x}$.*

*Proof.* $\|T(\boldsymbol{x})\|_2 = \|T(\boldsymbol{x}) - T(\boldsymbol{0})\|_2 \le \|\boldsymbol{x} - \boldsymbol{0}\|_2 = \|\boldsymbol{x}\|_2$. $\qquad\qquad\qquad\qquad\qquad\qquad\qquad\qquad\qquad\quad\square$

**Lemma A.2.** *Suppose for scales $s_i, s_j$ we have $f(\cdot|s_i) = G_{i,j} \circ f(\cdot|s_j)$ for some non-expansive $G_{i,j}$. Then, $\|f(\boldsymbol{x}|s_i)\|_2 \le \|f(\boldsymbol{x}|s_j)\|_2$. for all $\boldsymbol{x}$. Moreover, if there exists some $\boldsymbol{y}$ in the range of $f(\cdot|s_j)$ with $\|G_{i,j}(\boldsymbol{y})\|_2 < \|\boldsymbol{y}\|_2$, then there exists $\boldsymbol{x}' \in \mathcal{X}$ such that $\|f(\boldsymbol{x}'|s_i)\|_2 < \|f(\boldsymbol{x}'|s_j)\|_2$.*

*Proof.* By Lemma A.1, we have $\|G_{i,j}(\boldsymbol{u})\|_2 \le \|\boldsymbol{u}\|_2$ for all $\boldsymbol{u}$. Thus for any $\boldsymbol{x}$,

$$\|f(\boldsymbol{x}|s_i)\|_2 = \|G_{i,j}(f(\boldsymbol{x}|s_j))\|_2 \le \|f(\boldsymbol{x}|s_j)\|_2.$$

If some $\boldsymbol{y}$ in the range of $f(\cdot|s_j)$ satisfies $\|G_{i,j}(\boldsymbol{y})\|_2 < \|\boldsymbol{y}\|_2$, pick $\boldsymbol{x}'$ with $f(\boldsymbol{x}'|s_j) = \boldsymbol{y}$. Then

$$\|f(\boldsymbol{x}'|s_i)\|_2 = \|G_{i,j}(\boldsymbol{y})\|_2 < \|\boldsymbol{y}\|_2 = \|f(\boldsymbol{x}'|s_j)\|_2.$$

$\qquad\qquad\qquad\qquad\qquad\qquad\qquad\qquad\qquad\qquad\qquad\qquad\qquad\qquad\qquad\qquad\qquad\qquad\quad\square$

**Average Pooling.** Define

$$\left(f_{\mathrm{avg}}(\boldsymbol{x}|s)\right)_m := \frac{1}{s} \sum_{i \in B_m} x_i,$$

where $B_m = \{(m-1)s + 1, \dots, ms\}$ for $m = 1, \dots, L_s := P/s$.

*Non-expansiveness.* For any $\boldsymbol{x}, \boldsymbol{y}$ and any block $B_m$,

$$\left|f_{\mathrm{avg}}(\boldsymbol{x}|s)_m - f_{\mathrm{avg}}(\boldsymbol{y}|s)_m\right| = \frac{1}{s}\left|\sum_{i \in B_m}(x_i - y_i)\right|$$

$$\le \frac{1}{s}\sqrt{s}\,\|\boldsymbol{x}|_{B_m} - \boldsymbol{y}|_{B_m}\|_2 \le \|\boldsymbol{x}|_{B_m} - \boldsymbol{y}|_{B_m}\|_2.$$

Summing squares over $m$ yields

$$\|f_{\mathrm{avg}}(\boldsymbol{x}|s) - f_{\mathrm{avg}}(\boldsymbol{y}|s)\|_2 \le \|\boldsymbol{x} - \boldsymbol{y}\|_2,$$

so non-expansiveness holds.

*Energy Reduction.* Let $s_i = m \cdot s_j$ with $m > 1$. Each $s_i$-block is a union of $m$ consecutive $s_j$-blocks. Let $y = f_{\mathrm{avg}}(\boldsymbol{x}|s_j) \in \mathbb{R}^{L_{s_j}}$, and define

$$\left(G_{i,j}(y)\right)_k := \frac{1}{m}\sum_{r=0}^{m-1} y_{km+r},$$

i.e., average pooling on $y$ with block size $m$. Then one checks directly that

$$f_{\mathrm{avg}}(\boldsymbol{x}|s_i) = G_{i,j}\left(f_{\mathrm{avg}}(\boldsymbol{x}|s_j)\right),$$

and $G_{i,j}$ is the same blockwise averaging operator as above, hence non-expansive. By Lemma A.2, for all $\boldsymbol{x}$,

$$\|f_{\mathrm{avg}}(\boldsymbol{x}|s_i)\|_2 \le \|f_{\mathrm{avg}}(\boldsymbol{x}|s_j)\|_2.$$

To see strictness for some input, choose $\boldsymbol{y}$ with two distinct values in each $m$-block, so that the block average has strictly smaller energy than $\boldsymbol{y}$. Since $f_{\mathrm{avg}}(\cdot|s_j)$ is surjective onto $\mathbb{R}^{L_{s_j}}$, there exists $\boldsymbol{x}'$ with $f_{\mathrm{avg}}(\boldsymbol{x}'|s_j) = \boldsymbol{y}$, and Lemma A.2 yields

$$\|f_{\mathrm{avg}}(\boldsymbol{x}'|s_i)\|_2 < \|f_{\mathrm{avg}}(\boldsymbol{x}'|s_j)\|_2.$$

**Max- and Min-pooling.** Define

$$\left(f_{\max}(\boldsymbol{x}|s)\right)_m := \max_{i \in B_m} x_i, \quad \left(f_{\min}(\boldsymbol{x}|s)\right)_m := \min_{i \in B_m} x_i.$$

*Non-expansiveness.* For any $\boldsymbol{u}, \boldsymbol{v} \in \mathbb{R}^s$,

$$\left| \max_i u_i - \max_i v_i \right| \leq \max_i |u_i - v_i| \leq \|\boldsymbol{u} - \boldsymbol{v}\|_2,$$

and similarly for the minimum. Thus each block map is 1-Lipschitz; by summing over blocks we get

$$\|f_{\max}(\boldsymbol{x}|s) - f_{\max}(\boldsymbol{y}|s)\|_2 \leq \|\boldsymbol{x} - \boldsymbol{y}\|_2$$

and likewise for $f_{\min}$.

*Energy Reduction.* Let $s_i = m \cdot s_j$ with $m > 1$. Define $y = f_{\max}(\boldsymbol{x}|s_j)$ and

$$\left(G_{i,j}(y)\right)_k := \max_{r \in \{0, \cdots, m-1\}} y_{km+r},$$

i.e., max-pooling over $m$ consecutive entries of $y$. Then

$$f_{\max}(\boldsymbol{x}|s_i) = G_{i,j}\left(f_{\max}(\boldsymbol{x}|s_j)\right),$$

and the same max-Lipschitz argument as above shows $G_{i,j}$ is non-expansive. Thus by Lemma A.2, for all $\boldsymbol{x}$,

$$\|f_{\max}(\boldsymbol{x}|s_i)\|_2 \leq \|f_{\max}(\boldsymbol{x}|s_j)\|_2.$$

To get strictness, choose $\boldsymbol{y}$ such that in each group of $m$ entries there is at least one strictly smaller than the maximum; then $\|G_{i,j}(\boldsymbol{y})\|_2 < \|\boldsymbol{y}\|_2$. As before, we can realize such a $\boldsymbol{y}$ as $f_{\max}(\boldsymbol{x}'|s_j)$ by setting one large value and smaller ones within each block. The same reasoning applies to min-pooling by symmetry.

**Moving Average.** Define

$$\left(f_{\mathrm{ma}}(\boldsymbol{x}|s)\right)_t := \frac{1}{s} \sum_{i=0}^{s-1} x_{t-i}, \qquad t = 1, \ldots, P.$$

This is convolution with $h_s = \frac{1}{s}\mathbf{1}_s$.

*Non-expansiveness.* By Young's inequality,

$$\|f_{\mathrm{ma}}(\boldsymbol{x}|s) - f_{\mathrm{ma}}(\boldsymbol{y}|s)\|_2 = \|h_s * (\boldsymbol{x} - \boldsymbol{y})\|_2 \leq \|h_s\|_1 \|\boldsymbol{x} - \boldsymbol{y}\|_2 = \|\boldsymbol{x} - \boldsymbol{y}\|_2.$$

Thus $f_{\mathrm{ma}}(\cdot|s)$ is non-expansive.

*Energy Reduction.* Let $X(\omega)$ and $Y_s(\omega)$ denote the Fourier transforms of $\boldsymbol{x}$ and $f_{\mathrm{ma}}(\boldsymbol{x}|s)$, respectively, and $H_s(\omega)$ the frequency response of $h_s$. We have

$$Y_s(\omega) = H_s(\omega)X(\omega), \; H_s(\omega) = \frac{1}{s} \sum_{k=0}^{s-1} e^{-i\omega k} = \frac{1}{s} \frac{\sin(s\omega/2)}{\sin(\omega/2)} e^{-i\omega(s-1)/2}.$$

Thus

$$\|f_{\mathrm{ma}}(\boldsymbol{x}|s)\|_2^2 = \frac{1}{2\pi} \int_{-\pi}^{\pi} |H_s(\omega)|^2 |X(\omega)|^2 \, d\omega.$$

For $s_i = m \cdot s_j$, write $\alpha = s_j\omega/2$. Then

$$\frac{|H_{s_i}(\omega)|}{|H_{s_j}(\omega)|} = \frac{1}{m} \frac{|\sin(m\alpha)|}{|\sin(\alpha)|} \leq 1,$$

since $|\sin(m\alpha)| \le m|\sin(\alpha)|$ for all $m \in \mathbb{N}$ and all $\alpha$. Hence $|H_{s_i}(\omega)|^2 \le |H_{s_j}(\omega)|^2$ for all $\omega$, and therefore, for any $\boldsymbol{x}$,

$$\|f_{\mathrm{ma}}(\boldsymbol{x}|s_i)\|_2^2 \le \|f_{\mathrm{ma}}(\boldsymbol{x}|s_j)\|_2^2.$$

To see when the inequality is strict, define

$$D(\omega) := |H_{s_j}(\omega)|^2 - |H_{s_i}(\omega)|^2 \ge 0.$$

The above argument shows $D(\omega) \ge 0$ for all $\omega$, and $H_{s_i} \not\equiv H_{s_j}$ implies that $D(\omega) > 0$ on a set of nonzero measure. For any $\boldsymbol{x}$,

$$\|f_{\mathrm{ma}}(\boldsymbol{x}|s_j)\|_2^2 - \|f_{\mathrm{ma}}(\boldsymbol{x}|s_i)\|_2^2 = \frac{1}{2\pi} \int_{-\pi}^{\pi} D(\omega)\,|X(\omega)|^2\,d\omega.$$

Therefore,

$$\|f_{\mathrm{ma}}(\boldsymbol{x}|s_j)\|_2^2 > \|f_{\mathrm{ma}}(\boldsymbol{x}|s_i)\|_2^2$$

for every input $\boldsymbol{x}$ whose spectrum $|X(\omega)|^2$ places nonzero mass on the region where $D(\omega) > 0$. Equivalently, the inequality is strict for any non-degenerate $\boldsymbol{x}$ whose energy is not concentrated entirely on the set where $|H_{s_i}(\omega)| = |H_{s_j}(\omega)|$.

**Subsampling.** Define

$$\big(f_{\mathrm{sub}}(\boldsymbol{x}|s)\big)_m := x_{(m-1)s+1}, \qquad L_s = P/s.$$

*Non-expansiveness.* For each block $B_m$, define $g_s(\mathbf{u}) = u_1$. Then

$$|g_s(\mathbf{u}) - g_s(\mathbf{v})| = |u_1 - v_1| \le \|\mathbf{u} - \mathbf{v}\|_2.$$

Applying the blockwise arguments shows that $f_{\mathrm{sub}}(\cdot|s)$ is non-expansive.

*Energy Reduction.* If $s_i = m \cdot s_j$, pure decimation satisfies

$$f_{\mathrm{sub}}(\boldsymbol{x}|s_i) = f_{\mathrm{sub}}\big(f_{\mathrm{sub}}(\boldsymbol{x}|s_j)\big|m\big).$$

So $f_{\mathrm{sub}}(\cdot|s_i) = G_{i,j} \circ f_{\mathrm{sub}}(\cdot|s_j)$ with $G_{i,j}$ another subsampling operator, which is non-expansive. Hence Lemma A.2 yields

$$\|f_{\mathrm{sub}}(\boldsymbol{x}|s_i)\|_2 \le \|f_{\mathrm{sub}}(\boldsymbol{x}|s_j)\|_2$$

for all $\boldsymbol{x}$. The strict inequality holds by setting $\boldsymbol{x}$ as nonzero values concentrated on indices that are dropped at the coarser stride.

**Segmentation.** Define

$$f_{\mathrm{seg}}(\boldsymbol{x}|s) := (x_1, \ldots, x_{L_s}) \in \mathbb{R}^{L_s}, \qquad L_s = P/s,$$

*Non-expansiveness.* This is an orthogonal projection onto the first $L_s$ coordinates:

$$f_{\mathrm{seg}}(\boldsymbol{x}|s) = R_s \boldsymbol{x}, \quad R_s R_s^\top = I_{L_s}.$$

Thus

$$\|f_{\mathrm{seg}}(\boldsymbol{x}|s) - f_{\mathrm{seg}}(\boldsymbol{y}|s)\|_2 = \|R_s(\boldsymbol{x} - \boldsymbol{y})\|_2 \le \|\boldsymbol{x} - \boldsymbol{y}\|_2,$$

so $f_{\mathrm{seg}}$ is non-expansive and $\|f_{\mathrm{seg}}(\boldsymbol{x}|s)\|_2 \le \|\boldsymbol{x}\|_2$.

*Energy Reduction.* If $s_i = m \cdot s_j$, then

$$f_{\mathrm{seg}}(\boldsymbol{x}|s_i) = R_{s_i}\boldsymbol{x} = R_{s_i} R_{s_j}^\top f_{\mathrm{seg}}(\boldsymbol{x}|s_j).$$

Here $G_{i,j} := R_{s_i} R_{s_j}^\top$ is an orthogonal projection from $\mathbb{R}^{L_{s_j}}$ to $\mathbb{R}^{L_{s_i}}$, hence non-expansive. Thus by Lemma A.2, for all $\boldsymbol{x}$,

$$\|f_{\text{seg}}(\boldsymbol{x}|s_i)\|_2 \leq \|f_{\text{seg}}(\boldsymbol{x}|s_j)\|_2.$$

The inequality is strict exactly when at least one of the discarded coordinates is nonzero, i.e., when

$$x_k \neq 0 \quad \text{for some } k \in \{L_{s_i} + 1, \dots, L_{s_j}\}.$$

For such $\boldsymbol{x}$ we have

$$\sum_{k=L_{s_i}+1}^{L_{s_j}} x_k^2 > 0 \Rightarrow \|f_{\text{seg}}(\boldsymbol{x}|s_i)\|_2 < \|f_{\text{seg}}(\boldsymbol{x}|s_j)\|_2.$$

**Wavelet Decomposition.** Let $\{\phi_{s,k}\}_k$ be an orthonormal basis of $V_s$, and define the wavelet approximation coefficients by

$$f_{\text{wav}}(\boldsymbol{x}|s)_k := \langle \boldsymbol{x}, \phi_{s,k} \rangle.$$

For standard wavelets, the approximation spaces $\{V_s\}$ are typically constructed as nested subspaces indexed by dyadic scales $s = 2^j$. Here, we consider a more general multiplicative scale set $\mathcal{S} \subset \mathbb{Z}_+$ and assume that to each $s \in \mathcal{S}$ we associate an approximation subspace $V_s \subset \mathbb{R}^{L_s}$ with orthogonal projector $P_s : \mathbb{R}^L \to V_s$, such that

$$s_i = m s_j \text{ with } m > 1 \quad \Longrightarrow \quad V_{s_i} \subset V_{s_j} \text{ and } P_{s_i} = P_{s_i} P_{s_j}.$$

Equivalently, if $C_s : V_s \to \mathbb{R}^{L_s}$ denotes the isometry mapping $P_s \boldsymbol{x}$ to its coordinates in the basis $\{\phi_{s,k}\}_k$, then $f_{\text{wav}}(\boldsymbol{x}|s) = C_s P_s \boldsymbol{x} \in \mathbb{R}^{L_s}$.

*Non-expansiveness.* For any $s \in \mathcal{S}$, $P_s$ is an orthogonal projector, hence

$$\|P_s \boldsymbol{x} - P_s \boldsymbol{y}\|_2 \leq \|\boldsymbol{x} - \boldsymbol{y}\|_2.$$

Since $C_s$ is an isometry on $V_s$,

$$\|f_{\text{wav}}(\boldsymbol{x}|s) - f_{\text{wav}}(\boldsymbol{y}|s)\|_2 = \|C_s(P_s \boldsymbol{x} - P_s \boldsymbol{y})\|_2 = \|P_s \boldsymbol{x} - P_s \boldsymbol{y}\|_2 \leq \|\boldsymbol{x} - \boldsymbol{y}\|_2,$$

so $f_{\text{wav}}(\cdot|s)$ is non-expansive.

*Energy Reduction.* Fix $s_i, s_j \in \mathcal{S}$ with $s_i = m s_j, m > 1$. For any $\boldsymbol{x}$, we can write

$$f_{\text{wav}}(\boldsymbol{x}|s_j) = C_{s_j} P_{s_j} \boldsymbol{x}, \quad f_{\text{wav}}(\boldsymbol{x}|s_i) = C_{s_i} P_{s_i} \boldsymbol{x} = C_{s_i} P_{s_i} P_{s_j} \boldsymbol{x}.$$

Define a linear map $G_{i,j} : \mathbb{R}^{L_{s_j}} \to \mathbb{R}^{L_{s_i}}$ by

$$G_{i,j} := C_{s_i} P_{s_i} P_{s_j}^\top C_{s_j}^\top, \quad \text{so that} \quad f_{\text{wav}}(\boldsymbol{x}|s_i) = G_{i,j}(f_{\text{wav}}(\boldsymbol{x}|s_j)).$$

Here $C_{s_j}, C_{s_i}$ are isometries and $P_{s_i}$ is an orthogonal projector, so

$$\|G_{i,j}\|_2 \leq 1,$$

i.e., $G_{i,j}$ is non-expansive. By Lemma A.2, for all $\boldsymbol{x}$,

$$\|f_{\text{wav}}(\boldsymbol{x}|s_i)\|_2 = \|G_{i,j}(f_{\text{wav}}(\boldsymbol{x}|s_j))\|_2 \leq \|f_{\text{wav}}(\boldsymbol{x}|s_j)\|_2.$$

For strictness, pick any $\boldsymbol{x} \in V_{s_j} \setminus V_{s_i}$. Then $P_{s_j} \boldsymbol{x} = \boldsymbol{x}$, while $P_{s_i} \boldsymbol{x}$ is the orthogonal projection of $\boldsymbol{x}$ onto the strictly smaller subspace $V_{s_i}$, so by Pythagoras theorem, $\|P_{s_i} \boldsymbol{x}\|_2 < \|\boldsymbol{x}\|_2$. Then,

$$\|f_{\text{wav}}(\boldsymbol{x}|s_i)\|_2 = \|C_{s_i} P_{s_i} \boldsymbol{x}\|_2 < \|C_{s_j} \boldsymbol{x}\|_2 = \|f_{\text{wav}}(\boldsymbol{x}|s_j)\|_2.$$

## A.2. Proof of Theorem 3.3

*Proof.* We will show that each of operations is scale-degenerate in the sense that, for some pair $s_i > s_j$, we have

$$\|f(\boldsymbol{x}|s_i)\|_2 = \|f(\boldsymbol{x}|s_j)\|_2 \quad \text{for all } \boldsymbol{x} \in \mathcal{X},$$

so there exists no input $\boldsymbol{x}$ that yields strict inequality. In other words, all such families are degenerate in the scale parameter and therefore violate the energy reduction condition in Definition 3.1.

**Constant Mappings.** If $f(\boldsymbol{x}|s) = \boldsymbol{c}$ for some fixed $\boldsymbol{c} \in \mathbb{R}^{L_s}$, then

$$\|f(\boldsymbol{x}|s_i)\|_2 = \|f(\boldsymbol{x}|s_j)\|_2 = \|\boldsymbol{c}\|_2.$$

**Permutations.** If $f(\boldsymbol{x}|s) = \Pi_s \boldsymbol{x}$ for a permutation matrix $\Pi_s$, then

$$\|f(\boldsymbol{x}|s_i)\|_2 = \|f(\boldsymbol{x}|s_j)\|_2 = \|\boldsymbol{x}\|_2.$$

**Additive Shifts.** If $f(\boldsymbol{x}|s) = \boldsymbol{x} + \boldsymbol{b}$, then

$$\|f(\boldsymbol{x}|s_i)\|_2 = \|f(\boldsymbol{x}|s_j)\|_2 = \|\boldsymbol{x} + \boldsymbol{b}\|_2.$$

**Scalar Multiplications.** If $f(\boldsymbol{x}|s) = c\boldsymbol{x}$ for a constant $c \in \mathbb{R}$, then

$$\|f(\boldsymbol{x}|s_i)\|_2 = \|f(\boldsymbol{x}|s_j)\|_2 = |c|\|\boldsymbol{x}\|_2.$$

**General Linear Maps.** If $f(\boldsymbol{x}|s) = \boldsymbol{W}\boldsymbol{x}$, then

$$\|f(\boldsymbol{x}|s_i)\|_2 = \|f(\boldsymbol{x}|s_j)\|_2 = \|\boldsymbol{W}\boldsymbol{x}\|_2.$$

$\square$

## A.3. Proof of Theorem 4.2

*Proof.* Fix any $\boldsymbol{u} \in \Pi_1$. Consider

$$h_{\boldsymbol{u}}(\tau) := g_{\boldsymbol{x}}(t\mathbf{1} + \tau\boldsymbol{u}), \qquad \tau \in [0, 1].$$

By the fundamental theorem of calculus,

$$h_{\boldsymbol{u}}(\tau) = h_{\boldsymbol{u}}(0) + \tau h'_{\boldsymbol{u}}(0) + \int_0^\tau (\tau - r)\, h''_{\boldsymbol{u}}(r)\, dr.$$

Using $h''_{\boldsymbol{u}}(r) \leq \sup_{r' \in [0,\tau]} h''_{\boldsymbol{u}}(r')$ and $\int_0^\tau (\tau - r)\, dr = \tau^2/2$, we obtain for all $\tau \in [0, 1]$,

$$h_{\boldsymbol{u}}(\tau) \geq h_{\boldsymbol{u}}(0) + \tau h'_{\boldsymbol{u}}(0) - \frac{\tau^2}{2} \sup_{r \in [0,\tau]} h''_{\boldsymbol{u}}(r).$$

By the chain rule,

$$h'_{\boldsymbol{u}}(0) = \langle \nabla_s g_{\boldsymbol{x}}(t\mathbf{1}), \boldsymbol{u} \rangle, \qquad h''_{\boldsymbol{u}}(r) = \boldsymbol{u}^\top \nabla_s^2 g_{\boldsymbol{x}}(t\mathbf{1} + r\boldsymbol{u})\, \boldsymbol{u}.$$

Define the directional curvature envelope

$$\beta(\boldsymbol{x}) := \sup_{\boldsymbol{u} \in \Pi_1} \sup_{\tau \in [0,1]} \boldsymbol{u}^\top \nabla_s^2 g_{\boldsymbol{x}}(t\mathbf{1} + \tau\boldsymbol{u})\, \boldsymbol{u} \in (0, \infty).$$

Then

$$g_{\boldsymbol{x}}(t\mathbf{1} + \tau\boldsymbol{u}) \geq g_{\boldsymbol{x}}(t\mathbf{1}) + \tau \langle \nabla_s g_{\boldsymbol{x}}(t\mathbf{1}), \boldsymbol{u} \rangle - \frac{\beta(\boldsymbol{x})}{2}\tau^2, \qquad \forall \tau \in [0, 1].$$

For fixed $\boldsymbol{u}$, the right-hand side is a concave quadratic in $\tau$, whose maximum over $\tau \in [0,1]$ is at least

$$\frac{\langle \nabla_{\boldsymbol{s}} g_{\boldsymbol{x}}(t\mathbf{1}), \boldsymbol{u} \rangle^2}{2\beta(\boldsymbol{x})}.$$

Hence,

$$\max_{\tau \in [0,1]} g_{\boldsymbol{x}}(t\mathbf{1} + \tau \boldsymbol{u}) - g_{\boldsymbol{x}}(t\mathbf{1}) \geq C(\boldsymbol{x}) \langle \nabla_{\boldsymbol{s}} g_{\boldsymbol{x}}(t\mathbf{1}), \boldsymbol{u} \rangle^2,$$

where $C(\boldsymbol{x}) = 1/(2\beta(\boldsymbol{x}))$. Taking the supremum over $\boldsymbol{u} \in \Pi_1$, we obtain

$$\sup_{\boldsymbol{u} \in \Pi_1} \max_{\tau \in [0,1]} g_{\boldsymbol{x}}(t\mathbf{1} + \tau \boldsymbol{u}) - g_{\boldsymbol{x}}(t\mathbf{1}) \geq C(\boldsymbol{x}) \sup_{\boldsymbol{u} \in \Pi_1} \langle \nabla_{\boldsymbol{s}} g_{\boldsymbol{x}}(t\mathbf{1}), \boldsymbol{u} \rangle^2.$$

By definition, $\Phi_c(\boldsymbol{x}) = \max_{\boldsymbol{s} \in \mathcal{S}_c} g_{\boldsymbol{x}}(\boldsymbol{s})$. Since $\{t\mathbf{1} + \tau \boldsymbol{u} : \tau \in [0,1]\} \subset \mathcal{S}_c$ for all $\boldsymbol{u} \in \Pi_1$,

$$\Phi_c(\boldsymbol{x}) \geq \max_{\tau \in [0,1]} g_{\boldsymbol{x}}(t\mathbf{1} + \tau \boldsymbol{u}), \quad \forall \boldsymbol{u} \in \Pi_1.$$

Moreover, by construction of $t$ as a maximizer along the uniform-scale line,

$$g_{\boldsymbol{x}}(t\mathbf{1}) \geq \max_{s \in \mathcal{S}_d} g_{\boldsymbol{x}}(s\mathbf{1}) = \Phi_d(\boldsymbol{x}).$$

Combining the above inequalities yields

$$\Phi_c(\boldsymbol{x}) - \Phi_d(\boldsymbol{x}) \geq C(\boldsymbol{x}) \sup_{\boldsymbol{u} \in \Pi_1} \langle \nabla_{\boldsymbol{s}} g_{\boldsymbol{x}}(t\mathbf{1}), \boldsymbol{u} \rangle^2,$$

which proves the claim. $\qquad\square$

## A.4. Proof of Theorem 4.3

*Proof.* We verify (i) *differentiability* in $\boldsymbol{s}$ and (ii) *consistency* to a (discrete) scaling operator family.

**Differentiability in $\boldsymbol{s}$.** For each fixed pair $(i,j)$, the mapping $s_i \mapsto e^{-s_i} I_{|i-j|}(s_i)$ is real-analytic on $\mathbb{R}_+$ because $e^{-s_i}$ and the modified Bessel function $I_\nu(s_i)$ are analytic for $s_i > 0$ and integer $\nu$. Hence $\boldsymbol{s} \mapsto \boldsymbol{K}(\boldsymbol{s})$ is $C^\infty$ on $\mathbb{R}_+^M$, and so is $\boldsymbol{s} \mapsto k(\boldsymbol{x}|\boldsymbol{s}) = \boldsymbol{K}(\boldsymbol{s})\boldsymbol{x}$ for any $\boldsymbol{x}$. This proves the differentiability requirement in Theorem 4.1.

**Consistency to a Discrete Scaling Operator Family.** Fix $s \in \mathbb{Z}_+$ and set $\boldsymbol{s} = s\mathbf{1}$. Then $\boldsymbol{K}(s\mathbf{1})$ becomes space-invariant:

$$[\boldsymbol{K}(s\mathbf{1})]_{i,j} = e^{-s} I_{|i-j|}(s) =: g_s(i-j),$$

so $k(\boldsymbol{x}|s\mathbf{1})$ is the convolution of $\boldsymbol{x}$ with the kernel $g_s \in \ell^1(\mathbb{Z})$. Two classical properties hold:

*(a) Semigroup and normalization.* For all $s, t \geq 0$, $g_{s+t} = g_s * g_t$, and $\sum_{n \in \mathbb{Z}} g_s(n) = 1$. Thus $\boldsymbol{K}(s\mathbf{1})$ is a symmetric, doubly-stochastic Markov operator.

*(b) Frequency response and contraction.* Let $\widehat{g}_s(\omega)$ denote the discrete-time Fourier transform (DTFT) of $g_s$. A standard computation gives

$$\widehat{g}_s(\omega) = \exp\big(s(\cos\omega - 1)\big) = \exp\big(-2s\sin^2(\omega/2)\big), \qquad \omega \in [-\pi, \pi],$$

hence $|\widehat{g}_s(\omega)| \leq 1$ with equality only at $\omega = 0$ for $s > 0$.

We now verify the two axioms in Theorem 3.1 for the discrete family $f(\boldsymbol{x}|s) := k(\boldsymbol{x}|s\mathbf{1})$:

*Non-expansiveness.* By Plancherel,

$$\|f(\boldsymbol{x}|s) - f(\boldsymbol{x}'|s)\|_2^2 = \frac{1}{2\pi} \int_{-\pi}^{\pi} |\widehat{g}_s(\omega)|^2 \, |\widehat{\boldsymbol{x} - \boldsymbol{x}'}(\omega)|^2 \, d\omega \leq \|\boldsymbol{x} - \boldsymbol{x}'\|_2^2,$$

since $|\widehat{g}_s(\omega)| \leq 1$. Thus $\|f(\boldsymbol{x}|s) - f(\boldsymbol{x}'|s)\|_2 \leq \|\boldsymbol{x} - \boldsymbol{x}'\|_2$.

*Energy Reduction.* Let $s_i, s_j \in \mathbb{Z}_+$ with $s_i = m s_j$ for some integer $m > 1$. Let $X(\omega)$ denote the DTFT of $\boldsymbol{x}$; then by Plancherel,

$$\|f(\boldsymbol{x}|s)\|_2^2 = \frac{1}{2\pi} \int_{-\pi}^{\pi} |\widehat{g}_s(\omega)|^2 |X(\omega)|^2 \, d\omega.$$

Using the semigroup property,

$$g_{s_i} = g_{s_j}^{*m} := \underbrace{g_{s_j} * \cdots * g_{s_j}}_{m \text{ times}},$$

the frequency responses satisfy

$$\widehat{g}_{s_i}(\omega) = \left(\widehat{g}_{s_j}(\omega)\right)^m, \qquad |\widehat{g}_{s_i}(\omega)|^2 = |\widehat{g}_{s_j}(\omega)|^{2m}.$$

Since $|\widehat{g}_{s_j}(\omega)| \leq 1$ for all $\omega$ and $m > 1$, we have

$$|\widehat{g}_{s_i}(\omega)|^2 = |\widehat{g}_{s_j}(\omega)|^{2m} \leq |\widehat{g}_{s_j}(\omega)|^2 \quad \text{for all } \omega.$$

Therefore, for any $\boldsymbol{x}$,

$$\begin{aligned}
\|f(\boldsymbol{x}|s_i)\|_2^2 &= \frac{1}{2\pi} \int_{-\pi}^{\pi} |\widehat{g}_{s_i}(\omega)|^2 |X(\omega)|^2 \, d\omega \\
&\leq \frac{1}{2\pi} \int_{-\pi}^{\pi} |\widehat{g}_{s_j}(\omega)|^2 |X(\omega)|^2 \, d\omega \\
&= \|f(\boldsymbol{x}|s_j)\|_2^2,
\end{aligned}$$

which proves the *energy reduction* inequality

$$\|f(\boldsymbol{x}|s_i)\|_2 \leq \|f(\boldsymbol{x}|s_j)\|_2 \quad \text{for all } \boldsymbol{x} \quad \text{whenever } s_i = m s_j, \; m > 1.$$

To see that the inequality is strict for some $\boldsymbol{x}$, note that for $s_j > 0$ we have

$$|\widehat{g}_{s_j}(\omega)| < 1 \quad \text{for all } \omega \neq 0,$$

and hence

$$|\widehat{g}_{s_i}(\omega)|^2 = |\widehat{g}_{s_j}(\omega)|^{2m} < |\widehat{g}_{s_j}(\omega)|^2 \quad \text{for all } \omega \neq 0.$$

Choose any $\boldsymbol{x}$ whose spectrum is not supported solely at $\omega = 0$, i.e., such that $|X(\omega)|^2 > 0$ on a set of nonzero measure in $\{\omega \in [-\pi, \pi] : \omega \neq 0\}$. Then on a set of positive measure,

$$|\widehat{g}_{s_i}(\omega)|^2 |X(\omega)|^2 < |\widehat{g}_{s_j}(\omega)|^2 |X(\omega)|^2,$$

so the inequality above is strict:

$$\|f(\boldsymbol{x}|s_i)\|_2 < \|f(\boldsymbol{x}|s_j)\|_2.$$

Thus the LDG kernel family satisfies both the universal inequality and the existence of a strict example required by the energy reduction condition in Definition 3.1.

$\square$

## A.5. Proof of Theorem 4.4

### A.5.1. DISCRETE SCALE-SPACE AXIOMS

**Axioms (Discrete, 1D, Symmetric Case).** For each scale $s \geq 0$, let $\mathcal{T}_s : \ell^2(\mathbb{Z}) \to \ell^2(\mathbb{Z})$ denote the smoothing operator at scale $s$. A family $\{\mathcal{T}_s\}_{s \geq 0}$ satisfies the *discrete scale-space axioms* if:

(A1) **Linearity and Shift Invariance.** $\mathcal{T}_s$ is linear and commutes with shifts: there exists $K_s \in \ell^1(\mathbb{Z})$ with $(\mathcal{T}_s x)[n] = (K_s * x)[n]$.
(A2) **Semigroup over Scale and Identity at 0.** $\mathcal{T}_0 = \mathrm{Id}$ and $\mathcal{T}_{s+t} = \mathcal{T}_s \circ \mathcal{T}_t$, i.e., $K_{s+t} = K_s * K_t$ for all $s, t \geq 0$.
(A3) **Regularity in Scale.** The map $s \mapsto K_s$ is $C^1$ in $s$ in the $\ell^1$ topology.
(A4) **DC Normalization.** $\sum_{n \in \mathbb{Z}} K_s[n] = 1$ for all $s \geq 0$ (constants are preserved).
(A5) **Non-enhancement of Local Extrema (NELE).** If $y(\cdot; s) = \mathcal{T}_s x$ has a local maximum at index $n$, then $\partial_s y[n; s] \leq 0$ (resp. $\partial_s y[n; s] \geq 0$ at local minima).
(A6) **Symmetry.** $K_s[n] = K_s[-n]$ for all $n \in \mathbb{Z}$ and $s \geq 0$.

A.5.2. UNIQUENESS OF THE DISCRETE GAUSSIAN

**Lemma A.3.** *Let $A$ be a translation-invariant, symmetric operator on $\ell^2(\mathbb{Z})$ with impulse response $a[\cdot] \in \ell^1(\mathbb{Z})$ such that*

$$\sum_{n \in \mathbb{Z}} a[n] = 0, \qquad a[n] \geq 0 \text{ for } n \neq 0, \qquad a[-n] = a[n].$$

*Then its Fourier symbol $\phi(\omega) = \widehat{A}(\omega) = \sum_n a[n] e^{-in\omega}$ admits the Lévy-Khintchine form*

$$\phi(\omega) = \sum_{m=1}^{\infty} c_m \big( \cos(m\omega) - 1 \big), \qquad c_m \geq 0,$$

*with real coefficients $c_m = 2a[m]$. Conversely, given any sequence $\{c_m\}_{m \geq 1}$ with $c_m \geq 0$ and $\sum_{m \geq 1} c_m < \infty$, defining*

$$a[0] = -\sum_{m \geq 1} c_m, \qquad a[\pm m] = \tfrac{1}{2} c_m \ (m \geq 1),$$

*yields a symmetric Toeplitz operator $A$ with the above properties and symbol $\phi(\omega) = \sum_{m \geq 1} c_m (\cos m\omega - 1)$.*

*Proof.* Since $a \in \ell^1(\mathbb{Z})$ and $a[-n] = a[n]$,

$$\phi(\omega) = \sum_{n \in \mathbb{Z}} a[n] e^{-in\omega} = a[0] + 2 \sum_{m=1}^{\infty} a[m] \cos(m\omega).$$

The zero-sum condition gives $a[0] = -2 \sum_{m \geq 1} a[m]$, hence

$$\phi(\omega) = 2 \sum_{m=1}^{\infty} a[m] \big( \cos(m\omega) - 1 \big) = \sum_{m=1}^{\infty} c_m \big( \cos(m\omega) - 1 \big), \quad c_m := 2a[m] \geq 0.$$

Absolute summability of $a$ implies $\sum_m c_m < \infty$, ensuring uniform convergence and continuity of $\phi$. The converse follows by reversing the construction. $\square$

**Theorem A.4** (Uniqueness of the discrete Gaussian scale space). *Among symmetric kernel families, the* discrete Gaussian

$$K_s[n] = e^{-\alpha s} I_{|n|}(\alpha s), \qquad \alpha > 0,$$

*is the unique generalized scaling operator family that satisfies (A1)–(A6). Equivalently,*

$$\widehat{K}_s(\omega) = \exp\big(\alpha s(\cos \omega - 1)\big) = \exp\big(-2\alpha s \sin^2(\omega/2)\big), \qquad \omega \in [-\pi, \pi],$$

*and any other family satisfying the axioms coincides with $K_s$ up to reparameterizing scale $s \mapsto \alpha s$.*

*Proof.* By **(A1)**, $(\mathcal{T}_s x)[n] = (K_s * x)[n]$ for some $K_s \in \ell^1(\mathbb{Z})$. Let $\widehat{K}_s(\omega)$ be its DTFT. By **(A2)–(A3)**, for each fixed $\omega$ the map $s \mapsto \widehat{K}_s(\omega)$ is a continuous one-parameter semigroup with $\widehat{K}_0(\omega) = 1$, hence there exists a real, even *generator* $\phi(\omega)$ such that

$$\widehat{K}_s(\omega) = \exp\big(s\,\phi(\omega)\big), \qquad s \geq 0.$$

From **(A4)**, $\phi(0) = 0$. From **(A6)**, $\phi$ is real and even.

Let $y(\cdot; s) = \mathcal{T}_s x$. Differentiating in $s$,

$$\partial_s y = A * x, \qquad A := \partial_s K_s\big|_{s=0},$$

so that $\widehat{A}(\omega) = \phi(\omega)$. By **(A4)** and **(A6)**, $A$ is symmetric with zero row sum. The NELE axiom **(A5)** at vanishing scale enforces the discrete maximum principle for $A$: off-diagonal coefficients are nonnegative while the diagonal is nonpositive, and rows sum to zero. Thus $A$ fits the hypotheses of Theorem A.3, and

$$\phi(\omega) = \sum_{m=1}^{\infty} c_m \big( \cos(m\omega) - 1 \big), \qquad c_m \geq 0.$$

We now show that **(A5)** forces $c_m = 0$ for all $m \geq 2$. Consider signals supported on three consecutive sites and apply **(A5)** at $s = 0$ to both local maxima and minima. A standard extremum test yields that any positive coefficient at distance $m \geq 2$ would produce, for sufficiently small $s > 0$, an increase at a newly formed off-center extremum before nearest neighbors equilibrate. Hence $c_m = 0$ for $m \geq 2$, and

$$\phi(\omega) = c_1(\cos\omega - 1) =: \alpha(\cos\omega - 1), \qquad \alpha := c_1 > 0.$$

Substituting into equation A.5.2 gives

$$\widehat{K}_s(\omega) = \exp\big(\alpha s(\cos\omega - 1)\big).$$

Taking the inverse DTFT yields $K_s[n] = e^{-\alpha s} I_{|n|}(\alpha s)$, the discrete Gaussian kernel. This family satisfies **(A1)**–**(A6)**; conversely, any other family obeying the axioms has the same generator up to the multiplicative constant $\alpha$, i.e., a reparameterization of scale. $\qquad\square$

# B. Properties of Scaling Operator Family

Definition 3.1 characterizes a valid scaling operator through two essential properties: (1) non-expansiveness, requiring that applying the operator does not amplify differences between nearby inputs, and (2) energy reduction, requiring that coarser scales retain strictly less signal energy over multiplicable scales. These properties ensure that scaling progressively smooths the input while preserving stability.

Figure 8 reports the empirical behavior of all operator families in Table 1 across seven datasets. For every dataset, the induced output differences remain strictly smaller than the input differences, confirming non-expansiveness. Likewise, the average energy decreases monotonically over dyadic scales, indicating that each operator consistently removes high-frequency variation as the scale increases. Taken together, these results demonstrate that all operator families satisfy both conditions of Definition 3.1 not only in theory but also in practice, across diverse real-world time series.

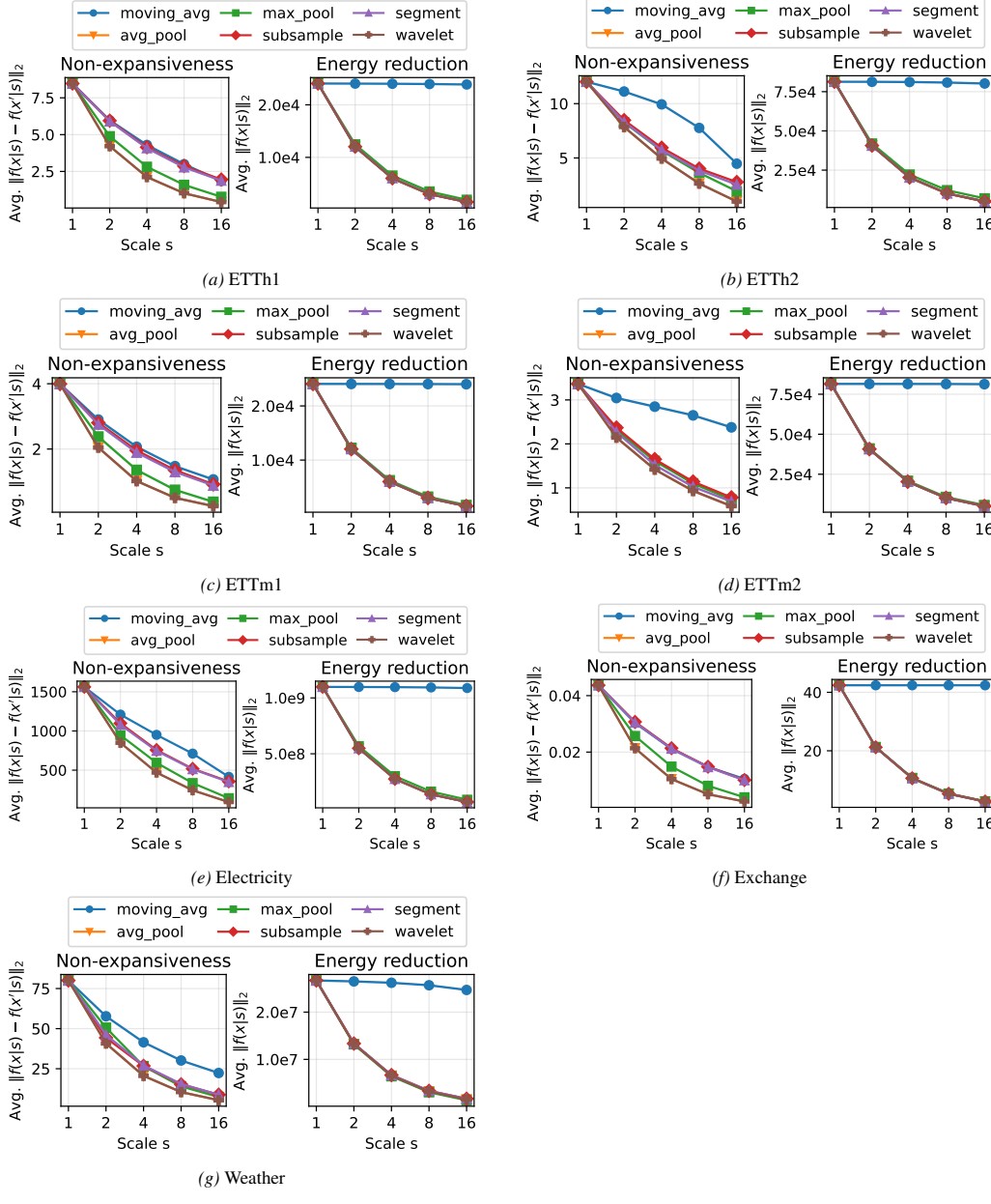

*Figure 8.* Energy reduction and non-expansiveness of the scaling operator families across all datasets.

# C. Empirical Validation of Theorem 4.2

We additionally provide empirical validation of Theorem 4.2 across all datasets. In this experiment, the scaling operator family $f(\mathbf{x}|\mathbf{s})$ is instantiated using an extended mean-pooling family, where integer scales recover standard pooling and continuous scales are obtained through interpolation. The results show that forecastability is consistently maximized at non-discrete scales and that the expressivity gap exceeds the theoretical lower bound across all samples.

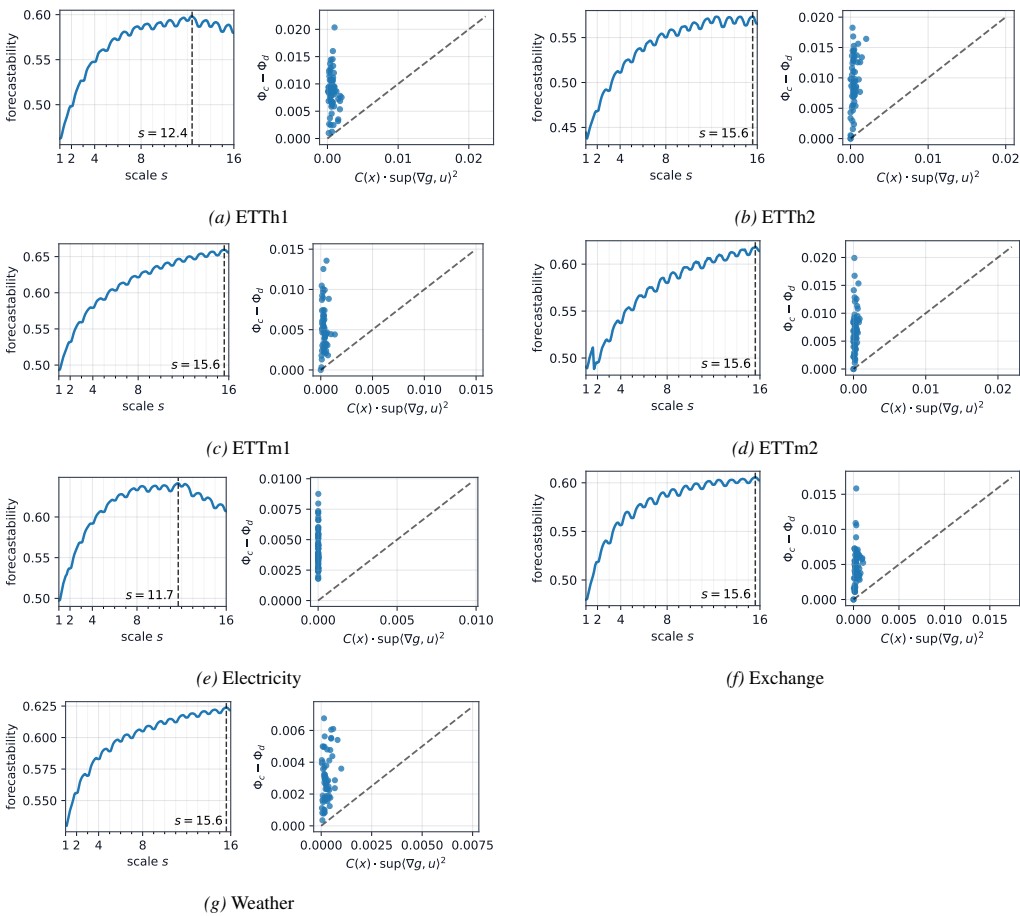

*(a)* ETTh1

*(b)* ETTh2

*(c)* ETTm1

*(d)* ETTm2

*(e)* Electricity

*(f)* Exchange

*(g)* Weather

*Figure 9.* Empirical validation of Theorem 4.2 across all datasets. The scaling operator $f$ is instantiated using an extended mean-pooling family. The left panels show that forecastability is consistently maximized at non-discrete scales, indicating $\Phi_c > \Phi_d$. The right panels show that the expressivity gap $\Phi_c - \Phi_d$ exceeds the theoretical lower bound for all samples. These results confirm that continuous scales yield strictly higher expressivity than discrete scales, supporting the theorem.

## D. Design Guide for Generalized Scaling Operator Families

The generalized scaling operator family proposed in Section 4 provides a flexible framework for constructing continuous, input-dependent multi-scale transformations. Here we summarize practical guidelines for designing new instances of this family and illustrate how these principles are instantiated in our implementation.

Classical downsampling operators are defined only at integer scales. To make these operators learnable and input-adaptive, we extend the discrete index $j \in \{1, \ldots, J\}$ to a continuous scale variable $s_t \in [1, J]$ at each position $t$. Instead of selecting a single integer operator, we blend nearby integer scales through a convex combination

$$f(\boldsymbol{x}|s_t) = \sum_{j=1}^{J} w_j(s_t) \, f_j(\boldsymbol{x})_t,$$

where $f_j$ is the original operator at scale $j$. This convex construction preserves non-expansiveness whenever each $f_j$ is non-expansive, and it ensures that $f(\cdot|s)$ varies smoothly with respect to $s$.

To satisfy Definition 3.1, the generalized operator must reproduce the original discrete operator exactly when $s_t$ is an integer. This is achieved by choosing weights $\{w_j(s)\}$ such that

$$w_j(k) = \mathbf{1}\{j = k\} \qquad \text{for all integers } k \in \{1, \ldots, J\}.$$

A convenient construction uses $C^\infty$ bump functions:

$$w_j(s) = \frac{\phi(s - j)}{\sum_{k=1}^{J} \phi(s - k)}, \qquad \phi(u) = \begin{cases} \exp\big(-1/(1 - u^2)\big), & |u| < 1, \\ 0, & \text{otherwise.} \end{cases}$$

At integer $s = k$, the only nonzero bump is $\phi(0)$, which yields $w_k(k) = 1$ and $w_{j \neq k}(k) = 0$. Thus the generalized operator reduces exactly to the classical operator at integer scales, ensuring consistency.

Because the bump function $\phi(\cdot)$ is $C^\infty$ and the normalization preserves smoothness, the weights $w_j(s)$ are differentiable in $s$ for all non-integer values. Since the output is a convex combination of the $f_j(\boldsymbol{x})$, the entire operator $f(\boldsymbol{x}|s)$ is differentiable in $s$, enabling backpropagation through the scale field and supporting input-adaptive scale prediction via a learnable scale head.

# E. Experimental Details

For the evaluation of forecasting models, we follow the protocol used in TimesNet (Wu et al., 2023). For **long-term forecasting** (see Table 2), we report the mean square error (MSE) and mean absolute error (MAE). For **short-term forecasting** (see Table 3), we adopt symmetric mean absolute percentage error (SMAPE), mean absolute scaled error (MASE), and overall weighted average (OWA). These metrics are defined as follows:

$$\text{SMAPE} = \frac{200}{H} \sum_{i=1}^{T} \frac{|\boldsymbol{X}_i - \hat{\boldsymbol{X}}_i|}{|\boldsymbol{X}_i| + |\hat{\boldsymbol{X}}_i|}, \tag{6}$$

$$\text{MASE} = \frac{1}{T} \sum_{i=1}^{T} \frac{|\boldsymbol{X}_i - \hat{\boldsymbol{X}}_i|}{\frac{1}{T-m} \sum_{j=m+1}^{T} |\boldsymbol{X}_j - \boldsymbol{X}_{j-m}|}, \tag{7}$$

$$\text{OWA} = \tfrac{1}{2} \left[ \frac{\text{SMAPE}}{\text{SMAPE}_{\text{Naive2}}} + \frac{\text{MASE}}{\text{MASE}_{\text{Naive2}}} \right], \tag{8}$$

where $m$ denotes the seasonal periodicity of the data, and $\boldsymbol{X} \in \mathbb{R}^{T \times C}$ and $\hat{\boldsymbol{X}} \in \mathbb{R}^{T \times C}$ represent the ground truth and predictions for $T$ future time steps with $C$ dimensions.

For methods that do not originally provide results on the M4 dataset, we use official implementations and conduct a controlled hyperparameter search for fair comparison. All models are implemented in PyTorch (Paszke et al., 2019), with the modified Bessel function implemented via SciPy, and all experiments are executed on a single **NVIDIA RTX A6000** GPU. Dataset statistics and detailed experimental configurations are provided in Tables 5 and 6, respectively.

*Table 5.* Dataset descriptions for long-term and short-term forecasting benchmarks. The dataset size is organized as (Train, Validation, Test). For long-term tasks, we adopt multivariate datasets covering diverse domains. For short-term tasks, we follow the official M4 benchmark, which consists of univariate series at different temporal frequencies. Following TimeMixer++ (Wang et al., 2025), forecastability is measured as one minus spectral entropy (Goerg, 2013), with higher values indicating better predictability.

| Tasks | Dataset | Dim | Series Length | Dataset Size | Domain | Frequency | Forecast. |
|---|---|---|---|---|---|---|---|
| | ETTm1 | 7 | {96, 192, 336, 720} | (34465, 11521, 11521) | Temperature | 15 min | 0.46 |
| | ETTm2 | 7 | {96, 192, 336, 720} | (34465, 11521, 11521) | Temperature | 15 min | 0.55 |
| | ETTh1 | 7 | {96, 192, 336, 720} | (8545, 2881, 2881) | Temperature | Hourly | 0.38 |
| Forecasting | ETTh2 | 7 | {96, 192, 336, 720} | (8545, 2881, 2881) | Temperature | Hourly | 0.45 |
| (Long-term) | Electricity | 321 | {96, 192, 336, 720} | (18317, 2633, 5261) | Electricity | Hourly | 0.77 |
| | Traffic | 862 | {96, 192, 336, 720} | (12185, 1757, 3509) | Transportation | Hourly | 0.68 |
| | Exchange | 8 | {96, 192, 336, 720} | (5120, 665, 1422) | Exchange rate | Daily | 0.41 |
| | Weather | 21 | {96, 192, 336, 720} | (36792, 5271, 10540) | Weather | 10 min | 0.75 |
| | M4-Yearly | 1 | 6 | (23000, 0, 23000) | Demographic | Yearly | 0.43 |
| | M4-Quarterly | 1 | 8 | (24000, 0, 24000) | Finance | Quarterly | 0.47 |
| Forecasting | M4-Monthly | 1 | 18 | (48000, 0, 48000) | Industry | Monthly | 0.44 |
| (Short-term) | M4-Weekly | 1 | 13 | (359, 0, 359) | Macro | Weekly | 0.43 |
| | M4-Daily | 1 | 14 | (4227, 0, 4227) | Micro | Daily | 0.44 |
| | M4-Hourly | 1 | 48 | (414, 0, 414) | Other | Hourly | 0.46 |

*Table 6.* Experiment configurations of SIGMA across datasets. All experiments adopt the ADAM (Kinga et al., 2015) optimizer. We report the model dimension ($d_{\text{model}}$), initial learning rate (LR), loss function, batch size, and training epochs.

| Dataset | $d_{\text{model}}$ | LR | Loss | Batch Size | Epochs |
|---|---|---|---|---|---|
| ETTh1 | 32 | 0.0005 | MSE | 32 | 10 |
| ETTh2 | 32 | 0.0005 | MSE | 32 | 10 |
| ETTm1 | 8 | 0.02 | MSE | 32 | 10 |
| ETTm2 | 16 | 0.0005 | MSE | 32 | 10 |
| Exchange | 16 | 0.0001 | MSE | 32 | 10 |
| Electricity | 16 | 0.01 | MSE | 32 | 10 |
| Traffic | 16 | 0.01 | MSE | 32 | 10 |
| Weather | 8 | 0.005 | MSE | 32 | 10 |
| M4 | 16 | 0.01 | SMAPE | 32 | 10 |

## F. Full Long-Term Forecasting Results

We report the full long-term forecasting results across all datasets and prediction horizons in Table 7. Overall, SIGMA achieves the best performance in 55 out of 80 settings and ranks second-best in 19 settings. The improvements are consistently observed across diverse datasets and forecasting horizons, demonstrating the effectiveness of principled multi-scale modeling for diverse long-horizon forecasting tasks.

*Table 7.* Full long-term forecasting results across eight datasets with horizons $T \in \{96, 192, 336, 720\}$ and input length fixed at 96. SIGMA achieves the smallest forecasting errors in 55 out of 80 evaluation settings and the second-best in 19 cases.

| Method | | SIGMA (Ours) | | AMD (2025a) | | MultiPatch. (2025) | | WPMixer (2025) | | TimeMixer (2024) | | MSGNet (2024) | | MICN (2023) | | TimesNet (2023) | | Pyra. (2022b) | |
|---|---|---|---|---|---|---|---|---|---|---|---|---|---|---|---|---|---|---|---|
| Metric | | MSE | MAE | MSE | MAE | MSE | MAE | MSE | MAE | MSE | MAE | MSE | MAE | MSE | MAE | MSE | MAE | MSE | MAE |
| Weather | 96 | **0.160** | **0.204** | 0.182 | 0.227 | 0.172 | 0.211 | 0.164 | 0.210 | 0.166 | 0.214 | 0.161 | 0.209 | 0.192 | 0.250 | 0.172 | 0.221 | 0.195 | 0.281 |
| | 192 | 0.209 | **0.248** | 0.231 | 0.266 | 0.218 | 0.254 | 0.209 | 0.250 | **0.208** | 0.251 | 0.217 | 0.257 | 0.233 | 0.289 | 0.225 | 0.265 | 0.245 | 0.322 |
| | 336 | 0.270 | 0.293 | 0.283 | 0.302 | 0.275 | 0.296 | **0.264** | **0.290** | 0.265 | 0.293 | 0.280 | 0.303 | 0.283 | 0.332 | 0.289 | 0.309 | 0.307 | 0.365 |
| | 720 | 0.348 | 0.345 | 0.357 | 0.350 | 0.355 | 0.348 | **0.344** | **0.343** | 0.345 | 0.345 | 0.373 | 0.362 | 0.354 | 0.388 | 0.361 | 0.355 | 0.394 | 0.420 |
| | Avg | 0.247 | **0.273** | 0.263 | 0.286 | 0.255 | 0.278 | **0.245** | **0.273** | 0.246 | 0.276 | 0.258 | 0.283 | 0.266 | 0.315 | 0.262 | 0.288 | 0.285 | 0.347 |
| Electricity | 96 | **0.146** | **0.241** | 0.187 | 0.269 | 0.173 | 0.259 | 0.167 | 0.259 | 0.157 | 0.248 | 0.169 | 0.281 | 0.171 | 0.284 | 0.165 | 0.268 | 0.283 | 0.377 |
| | 192 | **0.163** | **0.257** | 0.191 | 0.274 | 0.181 | 0.267 | 0.179 | 0.268 | 0.169 | 0.260 | 0.189 | 0.298 | 0.178 | 0.290 | 0.182 | 0.284 | 0.296 | 0.391 |
| | 336 | **0.179** | **0.273** | 0.206 | 0.290 | 0.199 | 0.285 | 0.197 | 0.288 | 0.187 | 0.277 | 0.201 | 0.310 | 0.189 | 0.301 | 0.198 | 0.299 | 0.306 | 0.401 |
| | 720 | 0.213 | **0.304** | 0.248 | 0.323 | 0.239 | 0.318 | 0.232 | 0.315 | 0.227 | 0.312 | 0.238 | 0.340 | **0.208** | 0.318 | 0.231 | 0.325 | 0.305 | 0.393 |
| | Avg | **0.175** | **0.269** | 0.208 | 0.289 | 0.198 | 0.282 | 0.194 | 0.282 | 0.185 | 0.274 | 0.199 | 0.307 | 0.187 | 0.298 | 0.194 | 0.294 | 0.298 | 0.390 |
| Traffic | 96 | **0.431** | **0.288** | 0.544 | 0.345 | 0.471 | 0.318 | 0.528 | 0.347 | 0.477 | 0.309 | 0.599 | 0.353 | 0.516 | 0.309 | 0.590 | 0.318 | 0.678 | 0.384 |
| | 192 | **0.444** | **0.296** | 0.527 | 0.335 | 0.480 | 0.319 | 0.511 | 0.337 | 0.488 | 0.312 | 0.634 | 0.372 | 0.535 | 0.317 | 0.614 | 0.327 | 0.672 | 0.377 |
| | 336 | **0.461** | **0.303** | 0.538 | 0.339 | 0.499 | 0.329 | 0.519 | 0.337 | 0.506 | 0.319 | 0.663 | 0.391 | 0.548 | 0.322 | 0.640 | 0.342 | 0.681 | 0.381 |
| | 720 | **0.494** | **0.320** | 0.573 | 0.358 | 0.537 | 0.351 | 0.548 | 0.350 | 0.535 | 0.332 | 0.721 | 0.420 | 0.574 | 0.332 | 0.662 | 0.350 | 0.709 | 0.395 |
| | Avg | **0.458** | **0.302** | 0.546 | 0.344 | 0.497 | 0.329 | 0.527 | 0.343 | 0.501 | 0.318 | 0.654 | 0.384 | 0.543 | 0.320 | 0.627 | 0.334 | 0.685 | 0.384 |
| Exchange | 96 | 0.084 | 0.204 | **0.083** | **0.201** | 0.089 | 0.208 | 0.086 | 0.202 | 0.090 | 0.210 | 0.104 | 0.230 | 0.093 | 0.226 | 0.112 | 0.242 | 0.630 | 0.645 |
| | 192 | **0.174** | 0.297 | 0.175 | 0.297 | 0.187 | 0.308 | 0.176 | **0.296** | 0.185 | 0.305 | 0.200 | 0.322 | 0.184 | 0.331 | 0.214 | 0.334 | 0.935 | 0.782 |
| | 336 | **0.322** | **0.411** | 0.326 | 0.412 | 0.357 | 0.436 | 0.343 | 0.421 | 0.351 | 0.428 | 0.394 | 0.460 | 0.322 | 0.443 | 0.379 | 0.452 | 1.204 | 0.873 |
| | 720 | 0.833 | **0.687** | 0.847 | 0.693 | 0.913 | 0.724 | 0.900 | 0.712 | 0.911 | 0.713 | 1.027 | 0.772 | **0.780** | 0.694 | 0.961 | 0.746 | 1.956 | 1.117 |
| | Avg | 0.353 | **0.400** | 0.358 | 0.401 | 0.387 | 0.419 | 0.376 | 0.408 | 0.384 | 0.414 | 0.431 | 0.446 | **0.345** | 0.424 | 0.416 | 0.443 | 1.181 | 0.854 |
| ETTh1 | 96 | 0.379 | **0.393** | 0.385 | 0.396 | **0.377** | 0.397 | 0.382 | 0.404 | 0.381 | 0.398 | 0.398 | 0.418 | 0.425 | 0.435 | 0.419 | 0.432 | 0.701 | 0.630 |
| | 192 | 0.430 | **0.425** | 0.437 | 0.425 | **0.427** | 0.428 | 0.438 | 0.427 | 0.441 | 0.434 | 0.444 | 0.445 | 0.505 | 0.484 | 0.474 | 0.464 | 0.850 | 0.713 |
| | 336 | 0.481 | 0.446 | 0.480 | **0.445** | **0.469** | 0.449 | 0.499 | 0.464 | 0.475 | 0.449 | 0.484 | 0.471 | 0.606 | 0.556 | 0.499 | 0.475 | 0.960 | 0.777 |
| | 720 | **0.480** | **0.468** | 0.485 | 0.469 | 0.499 | 0.485 | 0.487 | 0.471 | 0.522 | 0.494 | 0.509 | 0.498 | 0.752 | 0.648 | 0.529 | 0.500 | 1.005 | 0.803 |
| | Avg | **0.443** | **0.433** | 0.447 | 0.434 | **0.443** | 0.440 | 0.451 | 0.442 | 0.455 | 0.444 | 0.459 | 0.458 | 0.572 | 0.531 | 0.480 | 0.468 | 0.879 | 0.731 |
| ETTh2 | 96 | **0.289** | **0.339** | 0.291 | 0.340 | 0.293 | 0.347 | 0.291 | 0.342 | 0.296 | 0.348 | 0.327 | 0.369 | 0.358 | 0.405 | 0.327 | 0.369 | 1.439 | 0.922 |
| | 192 | **0.369** | **0.394** | 0.373 | 0.391 | 0.372 | 0.396 | 0.376 | 0.397 | 0.375 | 0.395 | 0.408 | 0.417 | 0.497 | 0.484 | 0.404 | 0.412 | 5.640 | 1.894 |
| | 336 | **0.415** | **0.427** | 0.416 | 0.427 | 0.420 | 0.431 | 0.435 | 0.439 | 0.430 | 0.438 | 0.429 | 0.439 | 0.618 | 0.552 | 0.459 | 0.456 | 4.800 | 1.844 |
| | 720 | 0.431 | 0.446 | **0.424** | **0.441** | 0.431 | 0.450 | 0.459 | 0.462 | 0.456 | 0.460 | 0.446 | 0.459 | 0.856 | 0.666 | 0.451 | 0.459 | 4.466 | 1.824 |
| | Avg | **0.376** | 0.402 | 0.376 | **0.400** | 0.379 | 0.406 | 0.390 | 0.410 | 0.389 | 0.410 | 0.402 | 0.421 | 0.582 | 0.527 | 0.410 | 0.424 | 4.086 | 1.621 |
| ETTm1 | 96 | 0.323 | 0.359 | 0.330 | 0.365 | **0.319** | **0.358** | 0.322 | **0.358** | 0.322 | 0.359 | 0.328 | 0.370 | 0.322 | 0.373 | 0.334 | 0.374 | 0.592 | 0.514 |
| | 192 | **0.360** | **0.381** | 0.372 | 0.384 | 0.363 | 0.385 | 0.361 | 0.382 | 0.364 | 0.384 | 0.371 | 0.395 | 0.361 | 0.402 | 0.404 | 0.409 | 0.645 | 0.568 |
| | 336 | 0.392 | 0.404 | 0.406 | 0.405 | 0.398 | 0.410 | **0.389** | **0.402** | 0.397 | 0.407 | 0.410 | 0.419 | 0.409 | 0.437 | 0.418 | 0.421 | 0.776 | 0.643 |
| | 720 | **0.455** | 0.442 | 0.471 | **0.440** | 0.460 | 0.448 | 0.469 | 0.447 | 0.456 | 0.444 | 0.494 | 0.465 | 0.506 | 0.498 | 0.484 | 0.455 | 0.936 | 0.730 |
| | Avg | **0.383** | **0.397** | 0.395 | 0.399 | 0.385 | 0.400 | 0.385 | 0.397 | 0.385 | 0.398 | 0.401 | 0.412 | 0.399 | 0.427 | 0.410 | 0.415 | 0.737 | 0.614 |
| ETTm2 | 96 | **0.174** | **0.257** | 0.183 | 0.267 | 0.177 | 0.259 | 0.175 | 0.257 | 0.176 | 0.258 | 0.179 | 0.263 | 0.186 | 0.283 | 0.187 | 0.266 | 0.387 | 0.464 |
| | 192 | **0.239** | **0.299** | 0.246 | 0.306 | 0.243 | 0.304 | 0.240 | 0.299 | 0.241 | 0.302 | 0.250 | 0.308 | 0.278 | 0.352 | 0.257 | 0.309 | 0.676 | 0.623 |
| | 336 | **0.296** | **0.337** | 0.305 | 0.342 | 0.305 | 0.346 | 0.304 | 0.342 | 0.304 | 0.345 | 0.311 | 0.345 | 0.405 | 0.437 | 0.322 | 0.349 | 1.196 | 0.836 |
| | 720 | **0.394** | **0.394** | 0.404 | 0.397 | 0.407 | 0.404 | 0.398 | 0.397 | 0.405 | 0.402 | 0.416 | 0.406 | 0.546 | 0.515 | 0.427 | 0.409 | 3.588 | 1.460 |
| | Avg | **0.276** | **0.322** | 0.285 | 0.328 | 0.283 | 0.328 | 0.279 | 0.324 | 0.281 | 0.327 | 0.289 | 0.330 | 0.354 | 0.397 | 0.298 | 0.333 | 1.462 | 0.846 |

## G. Efficiency Evaluation on Additional Datasets

We extend the efficiency analysis to more complex datasets, including Traffic and Electricity. As shown in Figure 10, the memory footprint of SIGMA increases with the number of time-series variables because of its channel-independent design. In contrast, methods such as AMD jointly model cross-variable dependencies, which can lead to lower memory usage in these cases. Nevertheless, SIGMA achieves substantially faster training due to its fully parallel formulation. In particular, it is 25.0× faster on Traffic and 22.5× faster on Electricity compared to AMD.

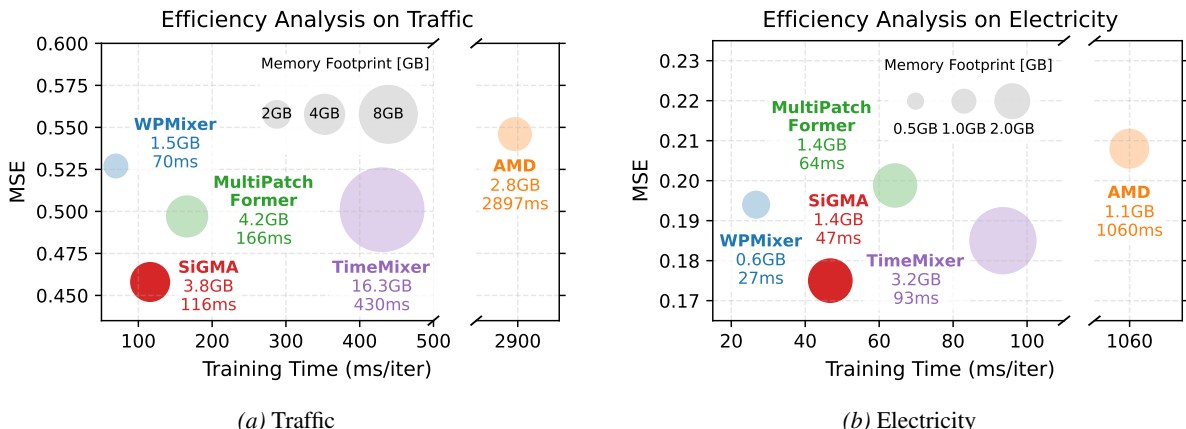

*(a)* Traffic                    *(b)* Electricity

*Figure 10.* Efficiency analysis on Traffic and Electricity with the predict-720 setting. SIGMA delivers the best accuracy while maintaining competitive training time and memory usage, showing robustness on more complex datasets.

## H. Computational Complexity and Efficient Application of LDG

The LDG operator adopts a distance-indexed parameterization, e.g., we learn a parameter vector $s \in \mathbb{R}^L$ indexed by pairwise distance $d = |i - j|$, and construct the kernel as

$$K_{ij} = e^{-s_d} I_d(s_d), \tag{9}$$

where $I_d(\cdot)$ denotes the modified Bessel function of the first kind. As a result, each entry depends only on the relative distance $d$, implying that the induced kernel matrix is symmetric Toeplitz. In practice, we compute the corresponding distance-aware kernel and apply the operator through matrix multiplication. More generally, the LDG operator can be interpreted as a one-dimensional convolution with kernel coefficients $K_d$, leading to several implementation regimes:

- **Dense matrix multiplication:** $O(L^2)$
- **Truncated convolution:** $O(LW)$
- **FFT-based Toeplitz multiplication:** $O(L \log L)$ (Kressner & Luce, 2018)

Here, $W$ denotes the effective kernel support. A standard definition based on tail mass is

$$W(\epsilon) = \min \left\{ w : \sum_{|d|>w} K_d \leq \epsilon \sum_d K_d \right\}, \tag{10}$$

where $\epsilon$ is a user-specified tolerance (Greengard & Strain, 1991).

We report the computational time and memory usage of different LDG implementations in Figure 11. The results show that truncated convolution exhibits the most favorable empirical scaling behavior, achieving approximately linear growth in both runtime and memory usage. By contrast, dense multiplication becomes increasingly expensive as $L$ grows, while FFT-based implementations incur substantial memory overhead and do not outperform simpler methods in the moderate-length regime. In practice, FFT-based implementations involve additional overhead from zero-padding to length $2L - 1$, complex-valued arithmetic, and multiple intermediate buffers (e.g., padded tensors and spectral-domain representations). Consequently, despite its favorable asymptotic complexity, FFT may be slower and more memory-intensive than simpler alternatives when $L$ is relatively small.

Overall, among the available strategies, truncated convolution provides the most favorable trade-off in practical settings, achieving near-linear scaling in both runtime and memory while effectively exploiting the strong locality induced by the LDG kernel.

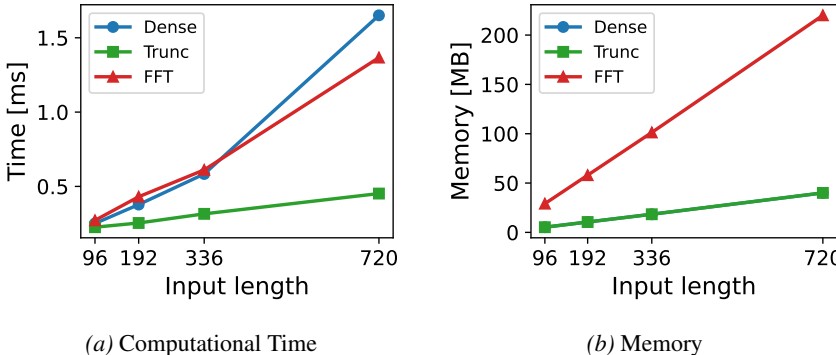

*(a)* Computational Time       *(b)* Memory

*Figure 11.* Empirical scaling analysis of the LDG operator under different implementation strategies. We report computational time (left) and memory usage (right) as functions of the input length $L$.

## I. Error Bars

To assess the robustness of our experiments, we report the mean performance and standard deviation for SIGMA and the second-best baselines in Tables 8 and 9. On the long-term forecasting benchmarks, SIGMA achieves lower average MSE and MAE than AMD on nearly all datasets, while maintaining sufficiently small standard deviations.

On the short-term M4 benchmark, the averaged SMAPE, MASE, and OWA scores of SIGMA remain consistently better, and the corresponding standard deviations are of similar or smaller magnitude. Taken together, these error-bar analyses confirm that the gains reported by SIGMA are stable across runs and not attributable to random fluctuations.

*Table 8.* Standard deviation for SIGMA and the second-best method (AMD) on long-term forecasting datasets.

| Method | SIGMA | | AMD (2025a) | |
|---|---|---|---|---|
| Dataset | MSE | MAE | MSE | MAE |
| Weather | $0.247 _{\pm 0.003}$ | $0.273 _{\pm 0.002}$ | $0.283 _{\pm 0.001}$ | $0.282 _{\pm 0.001}$ |
| Electricity | $0.175 _{\pm 0.001}$ | $0.269 _{\pm 0.001}$ | $0.208 _{\pm 0.000}$ | $0.303 _{\pm 0.000}$ |
| Traffic | $0.458 _{\pm 0.001}$ | $0.302 _{\pm 0.001}$ | $0.546 _{\pm 0.000}$ | $0.344 _{\pm 0.000}$ |
| Exchange | $0.353 _{\pm 0.001}$ | $0.400 _{\pm 0.001}$ | $0.358 _{\pm 0.009}$ | $0.401 _{\pm 0.002}$ |
| ETTh1 | $0.443 _{\pm 0.004}$ | $0.433 _{\pm 0.002}$ | $0.447 _{\pm 0.004}$ | $0.434 _{\pm 0.003}$ |
| ETTh2 | $0.376 _{\pm 0.003}$ | $0.402 _{\pm 0.002}$ | $0.376 _{\pm 0.022}$ | $0.400 _{\pm 0.005}$ |
| ETTm1 | $0.383 _{\pm 0.003}$ | $0.397 _{\pm 0.002}$ | $0.395 _{\pm 0.003}$ | $0.399 _{\pm 0.001}$ |
| ETTm2 | $0.276 _{\pm 0.001}$ | $0.322 _{\pm 0.001}$ | $0.285 _{\pm 0.043}$ | $0.328 _{\pm 0.012}$ |

*Table 9.* Standard deviation for SIGMA and the second-best method (MultiPatchFormer) on the short-term forecasting dataset (M4).

| Method | SIGMA | | | MultiPatchFormer (2025) | | |
|---|---|---|---|---|---|---|
| Dataset | SMAPE | MASE | OWA | SMAPE | MASE | OWA |
| Yearly | $13.314 _{\pm 0.022}$ | $2.989 _{\pm 0.015}$ | $0.783 _{\pm 0.002}$ | $13.296 _{\pm 0.012}$ | $3.009 _{\pm 0.008}$ | $0.785 _{\pm 0.001}$ |
| Quarterly | $10.060 _{\pm 0.052}$ | $1.177 _{\pm 0.009}$ | $0.886 _{\pm 0.005}$ | $10.166 _{\pm 0.008}$ | $1.178 _{\pm 0.003}$ | $0.892 _{\pm 0.002}$ |
| Monthly | $12.750 _{\pm 0.019}$ | $0.936 _{\pm 0.006}$ | $0.882 _{\pm 0.003}$ | $12.810 _{\pm 0.016}$ | $0.942 _{\pm 0.001}$ | $0.887 _{\pm 0.001}$ |
| Others | $4.867 _{\pm 0.046}$ | $3.316 _{\pm 0.050}$ | $1.037 _{\pm 0.011}$ | $4.849 _{\pm 0.037}$ | $3.271 _{\pm 0.023}$ | $1.028 _{\pm 0.007}$ |
| Averaged | $11.840 _{\pm 0.016}$ | $1.585 _{\pm 0.009}$ | $0.868 _{\pm 0.002}$ | $11.889 _{\pm 0.010}$ | $1.591 _{\pm 0.001}$ | $0.872 _{\pm 0.000}$ |

## J. Comparison with Recent Forecasting Baselines

We evaluate SIGMA against strong baselines, including TimePro (Ma et al., 2025), TimeFilter (Hu et al., 2025b), TimeMixer++ (Wang et al., 2025), iTransformer (Liu et al., 2024), PatchTST (Nie et al., 2023), and DLinear (Zeng et al., 2023) for long-term forecasting and short-term forecasting. Across the long-term forecasting benchmarks, SIGMA generally achieves the best accuracy on datasets with fewer variables (e.g., ETT). As the number of variables increases (e.g., Traffic), TimeFilter obtains the strongest overall performance, owing to its patch-wise filtration mechanism that explicitly captures spatiotemporal relationships. This suggests that explicitly modeling multi-scale patterns in multivariate settings can provide additional benefits. For short-term forecasting, SIGMA also performs strongly across all prediction horizons, demonstrating that SIGMA remains effective under diverse prediction-length settings.

*Table 10.* Long-term forecasting results comparing seven methods: SIGMA, TimePro, TimeFilter, TimeMixer++, iTransformer, PatchTST, and DLinear. Red and blue denote the best and second-best results, respectively.

| Method | | SIGMA (Ours) | | TimePro (2025) | | TimeFilter (2025b) | | TimeMixer++ (2025) | | iTransformer (2024) | | PatchTST (2023) | | DLinear (2023) | |
|---|---|---|---|---|---|---|---|---|---|---|---|---|---|---|---|
| Metric | | MSE | MAE | MSE | MAE | MSE | MAE | MSE | MAE | MSE | MAE | MSE | MAE | MSE | MAE |
| Weather | 96 | 0.160 | 0.204 | 0.174 | 0.213 | 0.159 | 0.205 | 0.163 | 0.211 | 0.176 | 0.216 | 0.174 | 0.216 | 0.196 | 0.258 |
| | 192 | 0.209 | 0.248 | 0.219 | 0.256 | 0.207 | 0.249 | 0.213 | 0.255 | 0.225 | 0.257 | 0.220 | 0.257 | 0.237 | 0.296 |
| | 336 | 0.270 | 0.293 | 0.276 | 0.299 | 0.264 | 0.291 | 0.273 | 0.298 | 0.281 | 0.299 | 0.278 | 0.298 | 0.283 | 0.333 |
| | 720 | 0.348 | 0.345 | 0.355 | 0.350 | 0.343 | 0.343 | 0.348 | 0.348 | 0.360 | 0.351 | 0.354 | 0.347 | 0.346 | 0.383 |
| | Avg | 0.247 | 0.273 | 0.256 | 0.279 | 0.243 | 0.272 | 0.249 | 0.278 | 0.260 | 0.281 | 0.256 | 0.279 | 0.266 | 0.318 |
| Electricity | 96 | 0.146 | 0.241 | 0.142 | 0.237 | 0.138 | 0.235 | 0.186 | 0.275 | 0.151 | 0.243 | 0.180 | 0.273 | 0.210 | 0.301 |
| | 192 | 0.163 | 0.257 | 0.160 | 0.252 | 0.156 | 0.253 | 0.190 | 0.277 | 0.167 | 0.258 | 0.187 | 0.279 | 0.210 | 0.305 |
| | 336 | 0.179 | 0.273 | 0.175 | 0.269 | 0.170 | 0.268 | 0.205 | 0.291 | 0.181 | 0.274 | 0.204 | 0.295 | 0.223 | 0.319 |
| | 720 | 0.213 | 0.304 | 0.212 | 0.302 | 0.191 | 0.289 | 0.249 | 0.327 | 0.213 | 0.301 | 0.246 | 0.328 | 0.258 | 0.350 |
| | Avg | 0.175 | 0.269 | 0.172 | 0.265 | 0.164 | 0.261 | 0.207 | 0.292 | 0.178 | 0.269 | 0.204 | 0.294 | 0.225 | 0.319 |
| Traffic | 96 | 0.431 | 0.288 | 0.400 | 0.267 | 0.391 | 0.260 | 0.570 | 0.365 | 0.397 | 0.271 | 0.460 | 0.298 | 0.696 | 0.429 |
| | 192 | 0.444 | 0.296 | 0.424 | 0.277 | 0.413 | 0.269 | 0.554 | 0.349 | 0.417 | 0.279 | 0.467 | 0.302 | 0.646 | 0.407 |
| | 336 | 0.461 | 0.303 | 0.443 | 0.286 | 0.429 | 0.277 | 0.568 | 0.352 | 0.432 | 0.287 | 0.483 | 0.308 | 0.653 | 0.410 |
| | 720 | 0.494 | 0.320 | 0.475 | 0.306 | 0.462 | 0.296 | 0.604 | 0.370 | 0.466 | 0.305 | 0.516 | 0.325 | 0.694 | 0.429 |
| | Avg | 0.458 | 0.302 | 0.435 | 0.284 | 0.424 | 0.276 | 0.574 | 0.359 | 0.428 | 0.285 | 0.482 | 0.308 | 0.672 | 0.419 |
| Exchange | 96 | 0.084 | 0.204 | 0.085 | 0.205 | 0.083 | 0.202 | 0.101 | 0.224 | 0.087 | 0.207 | 0.087 | 0.204 | 0.095 | 0.227 |
| | 192 | 0.174 | 0.297 | 0.179 | 0.301 | 0.178 | 0.299 | 0.200 | 0.321 | 0.179 | 0.302 | 0.188 | 0.308 | 0.184 | 0.323 |
| | 336 | 0.322 | 0.411 | 0.331 | 0.417 | 0.332 | 0.416 | 0.372 | 0.449 | 0.335 | 0.420 | 0.341 | 0.423 | 0.328 | 0.434 |
| | 720 | 0.833 | 0.687 | 0.921 | 0.724 | 0.785 | 0.669 | 1.000 | 0.762 | 0.854 | 0.697 | 0.921 | 0.720 | 0.762 | 0.667 |
| | Avg | 0.353 | 0.400 | 0.379 | 0.411 | 0.345 | 0.397 | 0.418 | 0.439 | 0.364 | 0.407 | 0.384 | 0.414 | 0.342 | 0.413 |
| ETTh1 | 96 | 0.379 | 0.393 | 0.378 | 0.399 | 0.384 | 0.395 | 0.407 | 0.418 | 0.388 | 0.405 | 0.379 | 0.398 | 0.396 | 0.410 |
| | 192 | 0.430 | 0.425 | 0.426 | 0.429 | 0.438 | 0.424 | 0.449 | 0.443 | 0.444 | 0.437 | 0.430 | 0.434 | 0.445 | 0.441 |
| | 336 | 0.481 | 0.446 | 0.469 | 0.452 | 0.481 | 0.446 | 0.493 | 0.466 | 0.486 | 0.457 | 0.473 | 0.460 | 0.493 | 0.471 |
| | 720 | 0.480 | 0.468 | 0.473 | 0.473 | 0.479 | 0.466 | 0.543 | 0.504 | 0.505 | 0.491 | 0.523 | 0.506 | 0.515 | 0.512 |
| | Avg | 0.443 | 0.433 | 0.436 | 0.438 | 0.446 | 0.433 | 0.473 | 0.458 | 0.456 | 0.448 | 0.451 | 0.450 | 0.462 | 0.459 |
| ETTh2 | 96 | 0.289 | 0.339 | 0.298 | 0.348 | 0.290 | 0.340 | 0.316 | 0.364 | 0.300 | 0.350 | 0.300 | 0.351 | 0.346 | 0.399 |
| | 192 | 0.369 | 0.394 | 0.367 | 0.395 | 0.377 | 0.394 | 0.402 | 0.418 | 0.379 | 0.398 | 0.380 | 0.400 | 0.478 | 0.477 |
| | 336 | 0.415 | 0.427 | 0.419 | 0.430 | 0.424 | 0.435 | 0.444 | 0.449 | 0.424 | 0.434 | 0.431 | 0.441 | 0.597 | 0.543 |
| | 720 | 0.431 | 0.446 | 0.429 | 0.446 | 0.464 | 0.464 | 0.464 | 0.469 | 0.433 | 0.448 | 0.447 | 0.461 | 0.841 | 0.661 |
| | Avg | 0.376 | 0.402 | 0.378 | 0.405 | 0.389 | 0.408 | 0.407 | 0.425 | 0.384 | 0.408 | 0.390 | 0.413 | 0.566 | 0.520 |
| ETTm1 | 96 | 0.323 | 0.359 | 0.328 | 0.366 | 0.320 | 0.357 | 0.332 | 0.368 | 0.344 | 0.377 | 0.330 | 0.368 | 0.345 | 0.372 |
| | 192 | 0.360 | 0.381 | 0.372 | 0.388 | 0.362 | 0.381 | 0.374 | 0.395 | 0.383 | 0.395 | 0.370 | 0.391 | 0.382 | 0.390 |
| | 336 | 0.392 | 0.404 | 0.401 | 0.408 | 0.391 | 0.402 | 0.386 | 0.407 | 0.417 | 0.418 | 0.402 | 0.411 | 0.414 | 0.414 |
| | 720 | 0.455 | 0.442 | 0.472 | 0.447 | 0.461 | 0.438 | 0.469 | 0.456 | 0.488 | 0.457 | 0.459 | 0.446 | 0.474 | 0.451 |
| | Avg | 0.383 | 0.397 | 0.393 | 0.402 | 0.383 | 0.394 | 0.390 | 0.407 | 0.408 | 0.412 | 0.390 | 0.404 | 0.404 | 0.407 |
| ETTm2 | 96 | 0.174 | 0.257 | 0.184 | 0.265 | 0.172 | 0.258 | 0.190 | 0.276 | 0.184 | 0.270 | 0.183 | 0.265 | 0.195 | 0.295 |
| | 192 | 0.239 | 0.299 | 0.245 | 0.306 | 0.237 | 0.300 | 0.256 | 0.315 | 0.251 | 0.312 | 0.246 | 0.308 | 0.282 | 0.359 |
| | 336 | 0.296 | 0.337 | 0.304 | 0.344 | 0.296 | 0.338 | 0.331 | 0.365 | 0.314 | 0.351 | 0.312 | 0.350 | 0.363 | 0.414 |
| | 720 | 0.394 | 0.394 | 0.405 | 0.402 | 0.394 | 0.396 | 0.429 | 0.420 | 0.412 | 0.406 | 0.419 | 0.412 | 0.547 | 0.519 |
| | Avg | 0.276 | 0.322 | 0.284 | 0.329 | 0.275 | 0.323 | 0.302 | 0.344 | 0.290 | 0.335 | 0.290 | 0.334 | 0.347 | 0.397 |

*Table 11.* Short-term forecasting results on the M4 benchmark across seven methods: SIGMA, TimePro, TimeFilter, TimeMixer++, iTransformer, PatchTST, and DLinear. Red and blue denote the best and second-best results, respectively. SIGMA achieves the best performance in 13 out of 15 evaluation cases, demonstrating consistently strong accuracy across temporal granularities and forecasting horizons.

| | Metric | SIGMA (Ours) | TimePro (2025) | TimeFilter (2025b) | TimeMixer++ (2025) | iTransformer (2024) | PatchTST (2023) | DLinear (2023) |
|---|---|---|---|---|---|---|---|---|
| Yearly | SMAPE | **13.314** | 13.368 | 18.836 | 13.368 | 14.091 | 14.311 | 14.343 |
| | MASE | **2.989** | 3.002 | 4.153 | 3.002 | 3.133 | 3.240 | 3.123 |
| | OWA | **0.783** | 0.787 | 1.099 | 0.787 | 0.825 | 0.845 | 0.832 |
| Quarterly | SMAPE | **10.060** | 10.121 | 10.660 | 10.269 | 11.775 | 10.242 | 10.502 |
| | MASE | **1.177** | 1.179 | 1.228 | 1.211 | 1.449 | 1.211 | 1.240 |
| | OWA | **0.886** | 0.890 | 0.932 | 0.908 | 1.063 | 0.907 | 0.929 |
| Monthly | SMAPE | **12.750** | 12.806 | 13.477 | 13.387 | 15.623 | 12.889 | 13.373 |
| | MASE | **0.936** | 0.940 | 1.025 | 1.019 | 1.273 | 0.955 | 1.004 |
| | OWA | **0.882** | 0.886 | 0.949 | 0.943 | 1.140 | 0.896 | 0.935 |
| Others | SMAPE | **4.867** | 5.326 | 6.136 | 5.535 | 5.407 | 4.986 | 5.110 |
| | MASE | 3.316 | 3.458 | 4.042 | 3.661 | 3.768 | **3.231** | 3.655 |
| | OWA | 1.037 | 1.081 | 1.257 | 1.160 | 1.163 | **1.023** | 1.132 |
| Weighted Average | SMAPE | **11.840** | 11.947 | 13.666 | 12.241 | 13.836 | 12.186 | 12.494 |
| | MASE | **1.585** | 1.614 | 1.944 | 1.653 | 1.868 | 1.656 | 1.681 |
| | OWA | **0.868** | 0.877 | 0.995 | 0.884 | 0.998 | 0.893 | 0.920 |

## K. Comparison Under Varying Look-Back Windows

To examine the effect of the look-back window beyond the fixed setting, we compare SiGMA and PatchTST over $L \in \{24, 48, 96, 192, 336, 512, 720\}$. Table 12 reports the best result for each dataset together with its selected configuration, while Tables 13 and 14 provide the full results across all look-back windows. Overall, SIGMA outperforms PatchTST on most datasets. In particular, on the Traffic dataset, SIGMA improves over PatchTST by 9.7% in MSE and 4.6% in MAE under the best-performing configurations. It also consistently outperforms PatchTST on all ETT benchmarks, suggesting that distance-aware multi-scale representations provide robust modeling capacity.

PatchTST also benefits from a broader search over context lengths. For example, it achieves its best performance on Weather with a relatively large input length ($L = 336$), whereas Exchange favors a much shorter context length ($L = 48$). These results suggest that the optimal look-back window is dataset-dependent rather than monotonically increasing with larger contexts. Moreover, increasing the look-back window incurs significantly higher cost for PatchTST. On Traffic with $L = 720$ under the predict-720 setting, one training epoch requires 4.0 minutes for SIGMA, compared to 23.3 minutes for PatchTST, making PatchTST approximately $5.8\times$ slower. Overall, SIGMA achieves robust forecasting performance across input lengths while maintaining substantially higher efficiency.

*Table 12.* Experimental evaluation against PatchTST over $L \in \{24, 48, 96, 192, 336, 512, 720\}$. For each dataset and method, we report the best performance across look-back windows, together with the look-back window $L$ that achieves it. Overall, the results show that SIGMA achieves robust forecasting performance across input lengths.

| Models | SIGMA | | | PatchTST | | |
|---|---|---|---|---|---|---|
| | MSE | MAE | $L$ | MSE | MAE | $L$ |
| Weather | 0.243 | 0.277 | 336 | **0.234** | **0.270** | 336 |
| Electricity | **0.163** | **0.259** | 192 | 0.165 | 0.266 | 720 |
| Traffic | **0.362** | **0.269** | 720 | 0.401 | 0.282 | 720 |
| Exchange | 0.353 | 0.400 | 96 | **0.351** | **0.397** | 48 |
| ETTh1 | **0.422** | **0.431** | 336 | 0.451 | 0.450 | 96 |
| ETTh2 | **0.366** | **0.400** | 336 | 0.386 | 0.415 | 192 |
| ETTm1 | **0.361** | **0.384** | 192 | 0.379 | 0.403 | 192 |
| ETTm2 | **0.266** | **0.320** | 336 | 0.287 | 0.337 | 192 |

*Table 13.* Long-term forecasting results SIGMA with different look-back windows.

| L | | $L=24$ | | $L=48$ | | $L=96$ | | $L=192$ | | $L=336$ | | $L=512$ | | $L=720$ | |
|---|---|---|---|---|---|---|---|---|---|---|---|---|---|---|---|---|
| Metric | | MSE | MAE | MSE | MAE | MSE | MAE | MSE | MAE | MSE | MAE | MSE | MAE | MSE | MAE |
| Weather | 96 | 0.197 | 0.229 | 0.185 | 0.223 | 0.160 | 0.204 | 0.164 | 0.213 | 0.162 | 0.216 | 0.161 | 0.216 | 0.161 | 0.214 |
| | 192 | 0.236 | 0.262 | 0.225 | 0.258 | 0.209 | 0.248 | 0.210 | 0.254 | 0.203 | 0.250 | 0.213 | 0.262 | 0.217 | 0.265 |
| | 336 | 0.299 | 0.307 | 0.282 | 0.299 | 0.270 | 0.293 | 0.262 | 0.291 | 0.263 | 0.295 | 0.277 | 0.306 | 0.291 | 0.318 |
| | 720 | 0.388 | 0.364 | 0.367 | 0.354 | 0.348 | 0.345 | 0.340 | 0.344 | 0.346 | 0.349 | 0.358 | 0.359 | 0.388 | 0.378 |
| | Avg | 0.280 | 0.290 | 0.265 | 0.283 | 0.247 | 0.272 | 0.244 | 0.276 | 0.243 | 0.277 | 0.252 | 0.286 | 0.264 | 0.294 |
| Electricity | 96 | 0.231 | 0.303 | 0.177 | 0.268 | 0.146 | 0.241 | 0.132 | 0.228 | 0.133 | 0.231 | 0.133 | 0.233 | 0.135 | 0.238 |
| | 192 | 0.233 | 0.307 | 0.189 | 0.277 | 0.163 | 0.257 | 0.154 | 0.249 | 0.160 | 0.259 | 0.158 | 0.258 | 0.158 | 0.259 |
| | 336 | 0.256 | 0.327 | 0.208 | 0.295 | 0.179 | 0.273 | 0.170 | 0.267 | 0.177 | 0.276 | 0.173 | 0.272 | 0.176 | 0.278 |
| | 720 | 0.300 | 0.358 | 0.242 | 0.325 | 0.213 | 0.304 | 0.195 | 0.291 | 0.190 | 0.289 | 0.198 | 0.297 | 0.198 | 0.299 |
| | Avg | 0.255 | 0.324 | 0.204 | 0.291 | 0.175 | 0.269 | 0.163 | 0.259 | 0.165 | 0.263 | 0.166 | 0.265 | 0.167 | 0.268 |
| Traffic | 96 | 0.599 | 0.371 | 0.514 | 0.332 | 0.431 | 0.288 | 0.380 | 0.264 | 0.357 | 0.259 | 0.341 | 0.255 | 0.337 | 0.254 |
| | 192 | 0.610 | 0.379 | 0.520 | 0.336 | 0.444 | 0.296 | 0.401 | 0.272 | 0.378 | 0.268 | 0.351 | 0.261 | 0.353 | 0.265 |
| | 336 | 0.635 | 0.387 | 0.544 | 0.347 | 0.461 | 0.303 | 0.416 | 0.280 | 0.381 | 0.275 | 0.366 | 0.268 | 0.360 | 0.270 |
| | 720 | 0.659 | 0.400 | 0.568 | 0.358 | 0.494 | 0.320 | 0.441 | 0.296 | 0.416 | 0.289 | 0.402 | 0.284 | 0.398 | 0.289 |
| | Avg | 0.626 | 0.384 | 0.537 | 0.343 | 0.458 | 0.302 | 0.410 | 0.278 | 0.383 | 0.272 | 0.365 | 0.267 | 0.362 | 0.269 |
| Exchange | 96 | 0.085 | 0.202 | 0.086 | 0.205 | 0.084 | 0.204 | 0.094 | 0.216 | 0.094 | 0.218 | 0.100 | 0.224 | 0.111 | 0.238 |
| | 192 | 0.187 | 0.306 | 0.187 | 0.308 | 0.174 | 0.297 | 0.204 | 0.323 | 0.195 | 0.316 | 0.210 | 0.329 | 0.224 | 0.343 |
| | 336 | 0.339 | 0.418 | 0.345 | 0.424 | 0.322 | 0.411 | 0.336 | 0.422 | 0.362 | 0.436 | 0.444 | 0.490 | 0.399 | 0.467 |
| | 720 | 0.888 | 0.710 | 0.915 | 0.720 | 0.833 | 0.687 | 1.068 | 0.765 | 1.043 | 0.766 | 1.554 | 0.896 | 1.375 | 0.881 |
| | Avg | 0.375 | 0.409 | 0.383 | 0.414 | 0.353 | 0.400 | 0.426 | 0.432 | 0.423 | 0.434 | 0.577 | 0.485 | 0.527 | 0.482 |
| ETTh1 | 96 | 0.430 | 0.416 | 0.385 | 0.395 | 0.379 | 0.393 | 0.388 | 0.403 | 0.378 | 0.398 | 0.374 | 0.401 | 0.386 | 0.412 |
| | 192 | 0.485 | 0.448 | 0.439 | 0.428 | 0.430 | 0.425 | 0.430 | 0.423 | 0.413 | 0.419 | 0.405 | 0.422 | 0.418 | 0.434 |
| | 336 | 0.538 | 0.476 | 0.490 | 0.453 | 0.481 | 0.446 | 0.458 | 0.438 | 0.437 | 0.436 | 0.434 | 0.440 | 0.454 | 0.456 |
| | 720 | 0.535 | 0.489 | 0.505 | 0.480 | 0.480 | 0.468 | 0.479 | 0.471 | 0.459 | 0.469 | 0.508 | 0.494 | 0.659 | 0.554 |
| | Avg | 0.497 | 0.457 | 0.455 | 0.439 | 0.443 | 0.433 | 0.439 | 0.434 | 0.422 | 0.431 | 0.430 | 0.439 | 0.479 | 0.464 |
| ETTh2 | 96 | 0.325 | 0.356 | 0.297 | 0.342 | 0.289 | 0.339 | 0.289 | 0.341 | 0.295 | 0.350 | 0.290 | 0.349 | 0.292 | 0.351 |
| | 192 | 0.416 | 0.407 | 0.383 | 0.394 | 0.369 | 0.394 | 0.367 | 0.394 | 0.355 | 0.389 | 0.364 | 0.394 | 0.373 | 0.407 |
| | 336 | 0.472 | 0.449 | 0.436 | 0.436 | 0.415 | 0.427 | 0.398 | 0.419 | 0.384 | 0.413 | 0.394 | 0.422 | 0.431 | 0.449 |
| | 720 | 0.477 | 0.461 | 0.454 | 0.455 | 0.431 | 0.446 | 0.435 | 0.451 | 0.431 | 0.450 | 0.449 | 0.464 | 0.472 | 0.478 |
| | Avg | 0.422 | 0.418 | 0.393 | 0.407 | 0.376 | 0.402 | 0.372 | 0.401 | 0.366 | 0.400 | 0.374 | 0.407 | 0.392 | 0.421 |
| ETTm1 | 96 | 0.677 | 0.502 | 0.467 | 0.426 | 0.323 | 0.359 | 0.300 | 0.346 | 0.304 | 0.350 | 0.306 | 0.357 | 0.319 | 0.364 |
| | 192 | 0.717 | 0.526 | 0.508 | 0.447 | 0.360 | 0.381 | 0.340 | 0.370 | 0.335 | 0.371 | 0.353 | 0.386 | 0.368 | 0.396 |
| | 336 | 0.761 | 0.551 | 0.551 | 0.473 | 0.392 | 0.404 | 0.368 | 0.391 | 0.374 | 0.397 | 0.377 | 0.402 | 0.414 | 0.425 |
| | 720 | 0.795 | 0.573 | 0.591 | 0.497 | 0.455 | 0.442 | 0.435 | 0.429 | 0.434 | 0.434 | 0.447 | 0.444 | 0.480 | 0.461 |
| | Avg | 0.737 | 0.538 | 0.529 | 0.461 | 0.383 | 0.397 | 0.361 | 0.384 | 0.362 | 0.388 | 0.370 | 0.397 | 0.395 | 0.411 |
| ETTm2 | 96 | 0.211 | 0.289 | 0.191 | 0.273 | 0.174 | 0.257 | 0.171 | 0.254 | 0.172 | 0.258 | 0.171 | 0.257 | 0.176 | 0.262 |
| | 192 | 0.280 | 0.331 | 0.259 | 0.316 | 0.239 | 0.299 | 0.234 | 0.296 | 0.239 | 0.301 | 0.246 | 0.306 | 0.240 | 0.303 |
| | 336 | 0.352 | 0.374 | 0.325 | 0.355 | 0.296 | 0.337 | 0.290 | 0.334 | 0.289 | 0.335 | 0.297 | 0.341 | 0.311 | 0.349 |
| | 720 | 0.458 | 0.431 | 0.428 | 0.412 | 0.394 | 0.394 | 0.381 | 0.392 | 0.365 | 0.386 | 0.374 | 0.392 | 0.385 | 0.401 |
| | Avg | 0.325 | 0.356 | 0.301 | 0.339 | 0.276 | 0.322 | 0.269 | 0.319 | 0.266 | 0.320 | 0.272 | 0.324 | 0.278 | 0.329 |

*Table 14.* Long-term forecasting results of PatchTST under different look-back windows. $L = 336$ and $L = 512$ correspond to PatchTST/42 and PatchTST/64, respectively. All settings use patch length $P = 16$ and stride $S = 8$.

| $L$ | | $L = 24$ | | $L = 48$ | | $L = 96$ | | $L = 192$ | | $L = 336$ | | $L = 512$ | | $L = 720$ | |
|---|---|---|---|---|---|---|---|---|---|---|---|---|---|---|---|
| Metric | | MSE | MAE | MSE | MAE | MSE | MAE | MSE | MAE | MSE | MAE | MSE | MAE | MSE | MAE |
| Weather | 96 | 0.220 | 0.247 | 0.208 | 0.242 | 0.174 | 0.216 | 0.157 | 0.204 | 0.153 | 0.203 | 0.151 | 0.204 | 0.158 | 0.214 |
| | 192 | 0.262 | 0.279 | 0.251 | 0.275 | 0.220 | 0.257 | 0.202 | 0.245 | 0.198 | 0.246 | 0.202 | 0.252 | 0.203 | 0.256 |
| | 336 | 0.323 | 0.322 | 0.306 | 0.313 | 0.278 | 0.298 | 0.259 | 0.288 | 0.254 | 0.289 | 0.262 | 0.301 | 0.260 | 0.301 |
| | 720 | 0.404 | 0.374 | 0.383 | 0.363 | 0.354 | 0.347 | 0.337 | 0.341 | 0.330 | 0.341 | 0.325 | 0.341 | 0.331 | 0.350 |
| | Avg | 0.302 | 0.305 | 0.287 | 0.298 | 0.257 | 0.279 | 0.239 | 0.270 | 0.234 | 0.270 | 0.235 | 0.274 | 0.238 | 0.280 |
| Electricity | 96 | 0.276 | 0.328 | 0.232 | 0.306 | 0.180 | 0.273 | 0.145 | 0.246 | 0.139 | 0.242 | 0.137 | 0.240 | 0.136 | 0.240 |
| | 192 | 0.268 | 0.329 | 0.227 | 0.306 | 0.187 | 0.279 | 0.161 | 0.261 | 0.156 | 0.257 | 0.153 | 0.255 | 0.153 | 0.255 |
| | 336 | 0.292 | 0.347 | 0.247 | 0.323 | 0.204 | 0.295 | 0.179 | 0.278 | 0.173 | 0.273 | 0.170 | 0.271 | 0.168 | 0.270 |
| | 720 | 0.330 | 0.377 | 0.287 | 0.354 | 0.246 | 0.328 | 0.217 | 0.310 | 0.210 | 0.304 | 0.205 | 0.300 | 0.202 | 0.298 |
| | Avg | 0.291 | 0.345 | 0.248 | 0.322 | 0.204 | 0.294 | 0.175 | 0.274 | 0.170 | 0.270 | 0.166 | 0.267 | 0.165 | 0.266 |
| Traffic | 96 | 0.753 | 0.416 | 0.651 | 0.375 | 0.460 | 0.298 | 0.400 | 0.277 | 0.386 | 0.273 | 0.379 | 0.273 | 0.374 | 0.270 |
| | 192 | 0.715 | 0.401 | 0.607 | 0.357 | 0.467 | 0.302 | 0.416 | 0.285 | 0.402 | 0.281 | 0.393 | 0.278 | 0.387 | 0.275 |
| | 336 | 0.748 | 0.414 | 0.631 | 0.366 | 0.483 | 0.308 | 0.430 | 0.290 | 0.413 | 0.286 | 0.404 | 0.283 | 0.402 | 0.282 |
| | 720 | 0.781 | 0.429 | 0.666 | 0.382 | 0.516 | 0.325 | 0.460 | 0.307 | 0.444 | 0.302 | 0.440 | 0.302 | 0.440 | 0.302 |
| | Avg | 0.749 | 0.415 | 0.639 | 0.370 | 0.482 | 0.308 | 0.427 | 0.290 | 0.411 | 0.286 | 0.404 | 0.284 | 0.401 | 0.282 |
| Exchange | 96 | 0.082 | 0.198 | 0.083 | 0.200 | 0.087 | 0.204 | 0.095 | 0.216 | 0.100 | 0.227 | 0.104 | 0.236 | 0.120 | 0.255 |
| | 192 | 0.170 | 0.292 | 0.174 | 0.294 | 0.188 | 0.308 | 0.212 | 0.329 | 0.236 | 0.351 | 0.249 | 0.365 | 0.267 | 0.378 |
| | 336 | 0.317 | 0.405 | 0.319 | 0.407 | 0.341 | 0.423 | 0.356 | 0.435 | 0.413 | 0.472 | 0.428 | 0.486 | 0.554 | 0.550 |
| | 720 | 0.841 | 0.690 | 0.827 | 0.685 | 0.921 | 0.720 | 0.890 | 0.699 | 1.058 | 0.757 | 1.110 | 0.779 | 1.361 | 0.863 |
| | Avg | 0.352 | 0.396 | 0.351 | 0.397 | 0.384 | 0.414 | 0.388 | 0.420 | 0.452 | 0.452 | 0.473 | 0.466 | 0.575 | 0.512 |
| ETTh1 | 96 | 0.415 | 0.416 | 0.384 | 0.402 | 0.379 | 0.398 | 0.423 | 0.437 | 0.451 | 0.452 | 0.455 | 0.464 | 0.470 | 0.470 |
| | 192 | 0.475 | 0.449 | 0.443 | 0.438 | 0.430 | 0.434 | 0.504 | 0.480 | 0.511 | 0.489 | 0.506 | 0.491 | 0.502 | 0.496 |
| | 336 | 0.530 | 0.481 | 0.498 | 0.471 | 0.473 | 0.460 | 0.531 | 0.498 | 0.552 | 0.513 | 0.520 | 0.509 | 0.560 | 0.535 |
| | 720 | 0.536 | 0.503 | 0.505 | 0.494 | 0.523 | 0.506 | 0.594 | 0.544 | 0.591 | 0.541 | 0.610 | 0.556 | 0.673 | 0.594 |
| | Avg | 0.489 | 0.462 | 0.457 | 0.451 | 0.451 | 0.450 | 0.513 | 0.490 | 0.526 | 0.499 | 0.523 | 0.505 | 0.551 | 0.524 |
| ETTh2 | 96 | 0.319 | 0.352 | 0.306 | 0.348 | 0.300 | 0.351 | 0.324 | 0.371 | 0.346 | 0.389 | 0.339 | 0.388 | 0.362 | 0.407 |
| | 192 | 0.413 | 0.406 | 0.391 | 0.399 | 0.380 | 0.400 | 0.384 | 0.408 | 0.452 | 0.450 | 0.443 | 0.443 | 0.441 | 0.443 |
| | 336 | 0.465 | 0.446 | 0.440 | 0.439 | 0.431 | 0.441 | 0.412 | 0.434 | 0.466 | 0.468 | 0.452 | 0.457 | 0.475 | 0.472 |
| | 720 | 0.470 | 0.461 | 0.448 | 0.456 | 0.447 | 0.461 | 0.425 | 0.445 | 0.457 | 0.473 | 0.508 | 0.500 | 0.491 | 0.490 |
| | Avg | 0.417 | 0.417 | 0.396 | 0.411 | 0.390 | 0.413 | 0.386 | 0.415 | 0.430 | 0.445 | 0.436 | 0.447 | 0.442 | 0.453 |
| ETTm1 | 96 | 0.635 | 0.486 | 0.424 | 0.404 | 0.330 | 0.368 | 0.324 | 0.369 | 0.324 | 0.374 | 0.336 | 0.385 | 0.329 | 0.384 |
| | 192 | 0.681 | 0.513 | 0.467 | 0.428 | 0.370 | 0.391 | 0.358 | 0.388 | 0.371 | 0.398 | 0.379 | 0.405 | 0.383 | 0.411 |
| | 336 | 0.727 | 0.539 | 0.506 | 0.453 | 0.402 | 0.411 | 0.389 | 0.412 | 0.400 | 0.420 | 0.417 | 0.429 | 0.454 | 0.459 |
| | 720 | 0.764 | 0.563 | 0.559 | 0.484 | 0.459 | 0.446 | 0.445 | 0.444 | 0.477 | 0.467 | 0.492 | 0.473 | 0.515 | 0.497 |
| | Avg | 0.702 | 0.525 | 0.489 | 0.442 | 0.390 | 0.404 | 0.379 | 0.403 | 0.393 | 0.415 | 0.406 | 0.423 | 0.420 | 0.438 |
| ETTm2 | 96 | 0.210 | 0.289 | 0.191 | 0.274 | 0.183 | 0.265 | 0.182 | 0.269 | 0.190 | 0.277 | 0.196 | 0.288 | 0.197 | 0.284 |
| | 192 | 0.281 | 0.333 | 0.261 | 0.319 | 0.246 | 0.308 | 0.244 | 0.312 | 0.249 | 0.319 | 0.253 | 0.325 | 0.264 | 0.331 |
| | 336 | 0.353 | 0.375 | 0.332 | 0.362 | 0.312 | 0.350 | 0.317 | 0.356 | 0.320 | 0.365 | 0.323 | 0.371 | 0.322 | 0.371 |
| | 720 | 0.457 | 0.432 | 0.432 | 0.418 | 0.419 | 0.412 | 0.404 | 0.410 | 0.420 | 0.428 | 0.423 | 0.427 | 0.406 | 0.422 |
| | Avg | 0.325 | 0.357 | 0.304 | 0.343 | 0.290 | 0.334 | 0.287 | 0.337 | 0.295 | 0.347 | 0.299 | 0.353 | 0.297 | 0.352 |

## L. Workflow Diagram and Algorithms

For completeness, we provide the overall workflow of SIGMA together with detailed algorithms for the end-to-end forward pass and the LDG kernel computations.

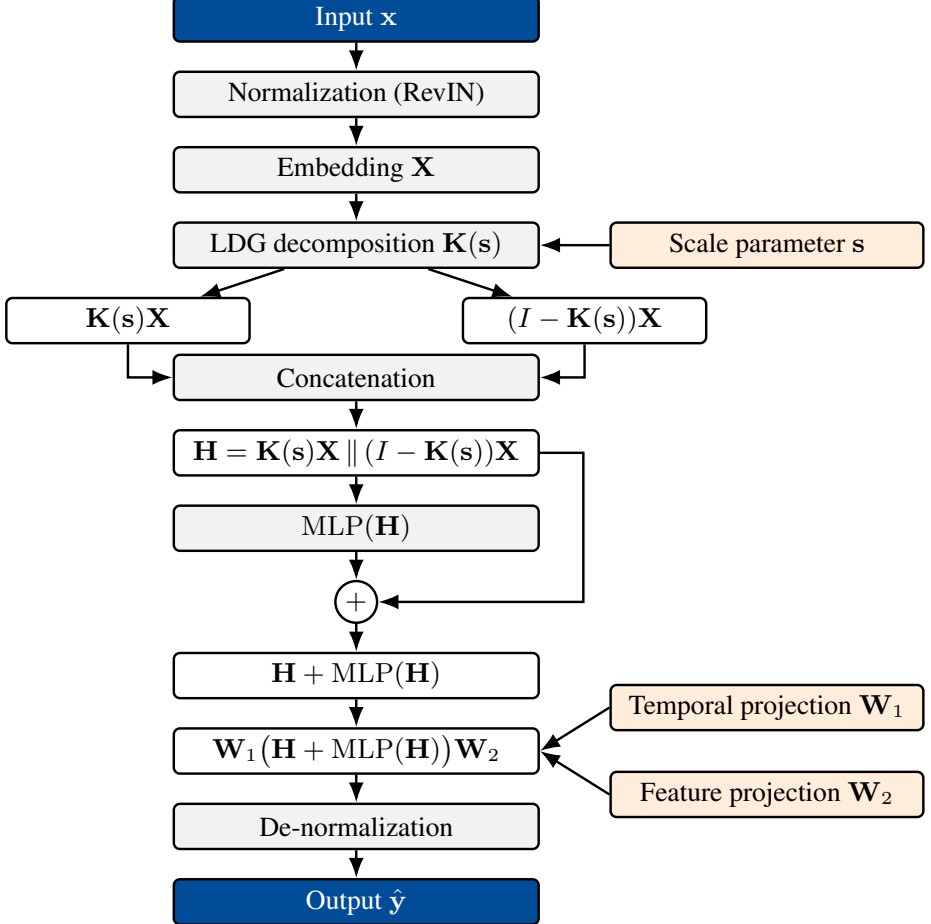

*Figure 12.* Workflow of SIGMA. The input $\mathbf{x}$ is normalized and embedded into $\mathbf{X}$. The LDG operator parameterized by $\mathbf{s}$ decomposes $\mathbf{X}$ into smoothed and residual components, which are concatenated into $\mathbf{H}$ and processed with a residual MLP. The resulting representation is projected to the prediction horizon via $\mathbf{W}_1$ and $\mathbf{W}_2$, followed by de-normalization to obtain $\hat{\mathbf{y}}$.

---

**Algorithm 1** Forward pass of SIGMA

---

1: **Input:** $\mathbf{x} \in \mathbb{R}^{B \times L \times C}$ with batch size $B$, look-back length $L$, and number of variables $C$
2: **Output:** $\hat{\mathbf{y}} \in \mathbb{R}^{B \times T \times C}$ with prediction horizon $T$
3: $\mathbf{x} \leftarrow \text{Normalize}(\mathbf{x})$
4: reshape $\mathbf{x} \in \mathbb{R}^{B \times L \times C}$ to $\mathbf{x} \in \mathbb{R}^{(BC) \times L \times 1}$
5: $\mathbf{H} \leftarrow \text{Embed}(\mathbf{x})$
6: $\mathbf{s} \leftarrow \text{softplus}(\boldsymbol{\theta})$
7: $(\mathbf{H}_s, \mathbf{H}_r) \leftarrow \text{LDGForward}(\mathbf{H}, \mathbf{s})$
8: $\bar{\mathbf{H}} \leftarrow \mathbf{H}_s \parallel \mathbf{H}_r$
9: $\mathbf{U} \leftarrow \bar{\mathbf{H}} + \text{MLP}(\bar{\mathbf{H}})$
10: split $\mathbf{U}$ into $\tilde{\mathbf{H}}_s, \tilde{\mathbf{H}}_r \in \mathbb{R}^{BC \times L \times d}$
11: $\mathbf{H} \leftarrow \tilde{\mathbf{H}}_s + \tilde{\mathbf{H}}_r$
12: **for** each branch $\mathbf{U} \in \{\tilde{\mathbf{H}}_s, \tilde{\mathbf{H}}_r\}$ **do**
13: $\quad \mathbf{Z} \leftarrow \mathbf{W}_1 \mathbf{U}$
14: $\quad \tilde{\mathbf{Y}}^{(\mathbf{U})} \leftarrow \mathbf{Z}\mathbf{W}_2$
15: $\quad$ reshape $\tilde{\mathbf{Y}}^{(\mathbf{U})}$ back to $\mathbb{R}^{B \times T \times C}$
16: **end for**
17: $\tilde{\mathbf{y}} \leftarrow \tilde{\mathbf{Y}}^{(\tilde{\mathbf{H}}_s)} + \tilde{\mathbf{Y}}^{(\tilde{\mathbf{H}}_r)}$
18: $\hat{\mathbf{y}} \leftarrow \text{Denormalize}(\tilde{\mathbf{y}})$
19: **return** $\hat{\mathbf{y}}$

---

**Algorithm 2** LDG kernel forward computation

---

1: **Input:** hidden representation $\mathbf{H} \in \mathbb{R}^{BC \times L \times d}$, scale parameters $\mathbf{s} \in \mathbb{R}^L_+$
2: **Output:** smoothed representation $\mathbf{H}_s \in \mathbb{R}^{BC \times L \times d}$, residual representation $\mathbf{H}_r \in \mathbb{R}^{BC \times L \times d}$
3: construct the distance matrix $\mathbf{D} \in \mathbb{N}^{L \times L}$ with $D_{i,j} = |i - j|$
4: **for** $d = 0, \ldots, L-1$ **do**
5: $\quad k_d \leftarrow e^{-s_d} I_d(s_d)$
6: **end for**
7: form the kernel matrix $\mathbf{K} \in \mathbb{R}^{L \times L}$ by $K_{i,j} = k_{D_{i,j}}$
8: $\mathbf{H}_s \leftarrow \mathbf{K}\mathbf{H}$
9: $\mathbf{H}_r \leftarrow \mathbf{H} - \mathbf{H}_s$
10: **return** $\mathbf{H}_s, \mathbf{H}_r$

---

**Algorithm 3** LDG kernel backward computation

---

1: **Input:** upstream gradient $\frac{\partial \mathcal{L}}{\partial \mathbf{K}} \in \mathbb{R}^{L \times L}$, scale parameters $\mathbf{s} \in \mathbb{R}^L_+$
2: **Output:** gradient w.r.t. $\mathbf{s}$, i.e., $\frac{\partial \mathcal{L}}{\partial \mathbf{s}} \in \mathbb{R}^L$
3: construct the distance matrix $\mathbf{D} \in \mathbb{N}^{L \times L}$ with $D_{i,j} = |i - j|$
4: **for** $d = 0, \ldots, L-1$ **do**
5: $\quad$ compute the derivative
$$\frac{\partial k_d}{\partial s_d} = e^{-s_d} \left( \frac{I_{d-1}(s_d) + I_{d+1}(s_d)}{2} - I_d(s_d) \right)$$
6: $\quad$ aggregate gradients for distance $d$:
$$g_d \leftarrow \sum_{i,j : D_{i,j} = d} \frac{\partial \mathcal{L}}{\partial K_{i,j}}$$
7: $\quad \frac{\partial \mathcal{L}}{\partial s_d} \leftarrow g_d \frac{\partial k_d}{\partial s_d}$
8: **end for**
9: **return** $\frac{\partial \mathcal{L}}{\partial \mathbf{s}}$

---

## M. Relation to Deformable Convolution or Pooling

SIGMA introduces adaptivity into multi-scale modeling by learning distance-aware scale parameters, rather than relying on fixed discrete scaling operators. Because SIGMA learns adaptive scale parameters, it is useful to distinguish it from other adaptive operators such as deformable convolution (Dai et al., 2017). While both approaches move beyond rigid uniform operators, their objectives are fundamentally different. Deformable convolution improves representational flexibility by adapting where information is sampled through learned input-dependent offsets, whereas SIGMA constructs a valid continuous-scale operator family for multi-scale representation learning.

Rather than modifying the sampling geometry, SIGMA controls how information is aggregated across temporal distances through a structured kernel. This distinction is important because the proposed operator satisfies principled properties, including non-expansiveness and energy reduction, thereby enabling consistent multi-scale behavior. Thus, SIGMA is best understood not as a deformable sampling mechanism, but as a structured continuous-scale operator for principled multi-scale time-series forecasting.

