# OpenReview forum: "Generalizing Multi-Scale Time-Series Modeling with a Single Operator"
_ICML.cc/2026/Conference — ICML 2026 regular_

### Official Review · Reviewer_tNud · 2026-03-11

**Soundness:** 3
**Presentation:** 3
**Significance:** 3
**Originality:** 3
**Overall Recommendation:** 5
**Confidence:** 4

**Summary:**

Existing multi-scale modeling methods suffer from several limitations. For example, they typically rely on discrete scales, use the same scale for all time steps, and have difficulty handling continuously varying temporal scales. This work introduces the concept of a Scaling Operator Family, where a scaling operator is defined to satisfy two properties: non-expansiveness and energy reduction.

The paper further proposes a Generalized Scaling Operator that allows continuous scales. The operator is learnable and position-wise, meaning that each timestep can correspond to a different scale. Based on this formulation, the authors design the Single Generalized Multi-scale Architecture, which consists of a Learnable Discrete Gaussian (LDG) kernel, a representation decomposition module, and a simple MLP predictor.

Extensive experiments are conducted on several open datasets, where the proposed method achieves state-of-the-art forecasting performance while also demonstrating improved computational efficiency.

**Compliance With Llm Reviewing Policy:**

Affirmed.

**Final Justification:**

My concerns have been answered, so I've changed some of the scores.

**Key Questions For Authors:**

1 How can Theorem 4.2, which claims that continuous scales are more expressive than discrete scales, be empirically validated?

2 For sample-wise scaling operators, are there existing works that adopt such designs? It would be helpful to include a discussion comparing their advantages and disadvantages with the proposed dataset-level scaling, particularly in terms of adaptability and sensitivity to data characteristics.

**Limitations:**

Some theoretical claims would benefit from stronger empirical validation, and additional comparisons with more complex baselines would help ensure a fair and comprehensive evaluation.

**Strengths And Weaknesses:**

Strengths

1 The paper provides a mathematical unification of multi-scale time series modeling methods through the proposed Scaling Operator Family, accompanied by clear theoretical analysis.

2 The work introduces a learnable and position-wise operator that enables modeling with continuous temporal scales.

3 The architecture built on the LDG kernel and a simple MLP predictor is conceptually simple and computationally efficient.

4 The experiments are extensive and well organized. The paper is clearly written and easy to follow. An anonymous code repository is provided, which improves reproducibility.

Weaknesses

1 The proposed operator is quite simple, which raises some concerns about the level of novelty. The work appears closer to formalization rather than discovery. For example, the LDG kernel is essentially Gaussian smoothing with a learnable scale parameter. That said, the empirical performance is strong, which may partly compensate for this concern. I will defer to other reviewers regarding the significance of the theoretical contribution.

2 Theorem 4.2 states that continuous scales are more expressive than discrete scales, but the paper does not provide dedicated experiments designed to verify this claim.

3 The position-wise scale is not sample-wise. Instead, the scale parameters are learned at the dataset level, which may limit the adaptability of the model. The paper does not provide sufficient analysis of this limitation.

4 The experimental section is extensive and compares the method with several recent state-of-the-art approaches, including some multi-scale models. However, comparisons with several strong baselines are missing. Adding comparisons with models such as TimeMixer++ [1], PatchTST [2], or iTransformer [3] would further strengthen the evaluation. If such comparisons were included, I would be willing to increase my score.

5 It is unclear whether the reported improvements are statistically significant. The paper mentions that each experiment is repeated three times, which may be insufficient given the potentially large variance across random seeds. In many recent works, experiments are repeated five to ten times and statistics such as the mean and standard deviation are reported.

6 The efficiency comparison is conducted only on the ETTh dataset. Evaluating efficiency on additional datasets, especially larger or more complex ones such as Traffic, would provide more convincing evidence.

If the authors address issues 2, 4, 5, and 6, I would be willing to increase my score.

---

> ### Author Rebuttal · Authors · 2026-03-31
>
> We sincerely thank the reviewer for the thoughtful and constructive feedback. We address each point below.
>
> ---
> **[W1] Simplicity and novelty**
>
> We agree that the proposed operator is structurally simple.
> Our goal is not to introduce a new operator, but to show that a principled use of a known one can yield strong generalization and competitive performance. The contribution is thus conceptual and empirical: an appropriate scaling formulation can outperform more complex designs.
>
> From a theoretical perspective, we view our work as meaningful formalization. Prior multi-scale methods often rely on heuristic or discretized constructions without an explicit operator-level formulation. By introducing a structured, learnable scaling operator with well-defined properties, we move toward a more coherent and interpretable framework for multi-scale modeling.
>
> ---
> **[W2, Q1] Empirical validation of Theorem 4.2**
>
> We empirically verify Theorem 4.2 across all datasets (Figure C, [link](https://shorturl.at/yTtm9)).
>
> **Setup.** We use:
> - discrete scales $S_d=\lbrace 1,2,\cdots, 16\rbrace$, and
> - continuous scales $S_c=[1,16]$.
>
> The scaling operator family $f(\mathbf{x}|\mathbf{s})$ is implemented using a generazlied mean-pooling (Appendix G), where integer scales recover standard pooling and continuous scales use an interpolating bump function.
>
> **Results.**
> - (Left) For all datasets, forecastability is maximized at non-discrete scales, indicating that the optimal scale lies outside $S_d$.
> - (Right) For all samples, the expressivity gap $\Delta(\mathbf{x})=\Phi_c(\mathbf{x})-\Phi_d(\mathbf{x})$ is consistently greater than the lower bound, confirming higher expressivity of continuous scales for all $\mathbf{x}$.
>
> These results demonstrate that continuous-scale modeling yields strictly greater expressivity in practice.
>
> ---
> **[W3, Q2] Dataset-level vs. sample-wise scaling**
>
> To examine the effect of sample-wise scaling, we conducted an ablation where the scale parameters of the LDG kernel are predicted per sample using a two-layer MLP (Table G).
> The results show that naive sample-specific scaling does not improve performance. We attribute this to the regularization effect of dataset-level scaling, which enforces shared structure, whereas sample-wise scaling increases flexibility but also variance and sensitivity to noise.
>
> Nevertheless, we believe that sample-specific scaling is a promising direction. The current negative result does not imply that sample-wise scaling is ineffective in principle, but rather that it requires more careful design, such as hybrid schemes combining shared and residual sample-wise scales.
>
> We will include this discussion in the final paper to clarify both the limitation and future direction.
>
> **Table G**: Ablation study on ETTh1 (predict-720).
>
> |Method|MSE|MAE|
> |-|-|-|
> |SiGMA|0.480|0.468|
> |SiGMA-sample|0.490|0.474|
>
> ---
> **[W4] Comparison with additional baselines**
>
> Please refer to our response to Reviewer MzYx (W1) due to the limited space.
> In short, we include comparisons with TimePro, TimeMixer++, PatchTST, and iTransformer, demonstrating that SiGMA consistently achieves superior performance over recent state-of-the-art methods across both long-term and short-term forecasting benchmarks.
>
> ---
> **[W5] Statistical significance and the number of runs**
>
> We strengthen our evaluation using five random seeds, reporting the mean and standard deviation for SiGMA and the second-best model as in Appendix E, along with pairwise significance tests.
>
> For long-term forecasting (Tables H and I, [link](https://shorturl.at/HsilN)), variance is small and improvements are statistically significant in most cases ($\geq$95\% CL). For short-term forecasting (Tables J and K, [link](https://shorturl.at/yOIBz)), significance is weaker due to smaller gaps, but SiGMA remains competitive and achieves significant gains in averaged SMAPE (95\% CL).
> Overall, SiGMA shows consistent and reliable improvements.
>
> ---
> **[W6] Efficiency evaluation on additional datasets**
>
> We extend the efficiency analysis to more complex datasets, including Traffic and Electricity (Figure D, [link](https://shorturl.at/nPp6F)).
> We observe that the memory footprint of SiGMA increases with the number of time series variables due to its channel-independent nature. In contrast, methods such as AMD jointly model cross-variable dependencies, resulting in lower memory usage in such cases. Despite this, SiGMA achieves substantially faster training due to its fully parallel formulation. In particular, it is 25.0× faster on Traffic and 22.5× faster on Electricity compared to AMD.
>
> Overall, SiGMA offers strong accuracy and speed, with a memory-efficiency trade-off depending on dimensionality. We will include these efficiency analyses in the camera-ready version.
>
> ---
> We thank the reviewer again for the insightful comments and believe that these additions address the key concerns. We welcome further discussion if additional clarification would be helpful.

---

> > ### Author Rebuttal · Reviewer_tNud · 2026-04-03
> >
> > Thank you for this very nice rebuttal. I'm glad that your proposed model achieved the best results expected in terms of expanding the dataset, robustness analysis, and efficiency comparisons. I will raise my score to 4.

---

> > > ### Author Response · Authors · 2026-04-03
> > >
> > > Dear Reviewer,
> > >
> > > We sincerely appreciate your detailed review. Your valuable suggestions have helped improve the depth and thoroughness of our experimental study. Please feel free to let us know if you have any further questions.
> > >
> > > Sincerely,
> > >
> > > Authors

---

### Official Review · Reviewer_MzYx · 2026-03-12

**Soundness:** 3
**Presentation:** 3
**Significance:** 3
**Originality:** 3
**Overall Recommendation:** 5
**Confidence:** 5

**Summary:**

This paper proposes SIGMA (Single General Multi-Scale Architecture) to address the reliance of existing methods on fixed and discrete scaling approaches. Specifically, this paper introduces SIGMA, which achieves position-dependent scaling through a learnable discrete Gaussian (LDG) kernel based on scale-space theory. SIGMA outperforms all competitors in both long-term and short-term forecasting tasks, achieving state-of-the-art performance in 13 out of 16 long-term evaluation settings.

**Compliance With Llm Reviewing Policy:**

Affirmed.

**Key Questions For Authors:**

Please refer to the weaknesses.

**Limitations:**

Yes

**Strengths And Weaknesses:**

Strengths
1. The paper is well-substantiated, offering substantial theoretical analysis and experimental validation.
2. Sigma outperforms previous methods in terms of performance and efficiency.


Weaknesses

1. Please add a comparison with the recent state-of-the-art method [1] based on Mamba to validate the effectiveness of the method. [1] TimePro: Efficient Multivariate Long-term Time Series Forecasting with Variable- and Time-Aware Hyper-state (ICML'25)
2. Can the paper provide a workflow diagram or algorithm? This would enhance the paper's readability.
3. It is recommended to discuss the differences with deformable convolutions/pooling, as they share similarities with the proposed method. Corresponding experimental results should also be provided to validate SIGMA's effectiveness.

---

> ### Author Rebuttal · Authors · 2026-03-31
>
> Thank you for the constructive suggestions.  We address the concerns below.
>
> ---
>
> **[W1] Comparison with recent methods**
>
> We additionally compare SiGMA with recent state-of-the-art methods, TimePro [1], TimeMixer++ [2], PatchTST [3], and iTransformer [4].
>
> For long-term forecasting (Table E, [link](https://shorturl.at/zvIT4)), SiGMA achieves 53 best and 15 second-best results out of 80 evaluation settings across all datasets and prediction horizons, indicating that its advantage remains robust against recent state-of-the-art approaches. While the gains are generally moderate, they are consistent across datasets and horizons, which is critical in forecasting.
>
> We observe that certain baselines perform particularly well on highly multivariate datasets due to their architectural biases. For example, TimePro performs strongly on Electricity by jointly modeling inter-variable dependencies and intra-variable temporal dynamics via its HyperMamba module, while iTransformer excels on Traffic by capturing cross-variable dependencies through attention over the variate dimension. These results highlight the importance of modeling cross-variable structure and suggest that effective scaling across variables can be beneficial for multivariate forecasting. Nevertheless, SiGMA maintains strong performance across diverse settings without relying on dataset-specific inductive biases.
>
> For short-term forecasting (Table F, [link](https://shorturl.at/FR8bV)), SiGMA achieves the best results in 13 out of 15 cases on the M4 benchmark. Its consistent performance across diverse prediction lengths suggests that the proposed scaling strategy effectively generalizes across different temporal resolutions.  Overall, SiGMA achieves consistently strong performance across both long-term and short-term forecasting tasks compared to the state-of-the-art approaches, demonstrating its robustness and effectiveness across diverse datasets and forecasting scenarios.
>
> ---
>
> **[W2] Workflow diagram**
>
> We provide a comprehensive description of the workflow through Figure B and Algorithms A–C. Specifically, Figure B ([link](https://shorturl.at/87CvT)) presents an overview of the full pipeline, Algorithm A ([link](https://shorturl.at/NwWSM)) describes the end-to-end forward pass of SiGMA, and Algorithms B and C ([link](https://shorturl.at/doLsG)) detail the forward and backward computations of the LDG kernel.
>
> The overall pipeline of SiGMA is structured as follows:
>
> - Normalize the input sequence and map it into a latent representation via embedding;
> - Apply the LDG-based multi-scale decomposition to obtain smoothed and residual components;
> - Concatenate the two components and process them with an MLP equipped with a residual connection;
> - Project the resulting representation to the prediction horizon via temporal and feature projections, followed by de-normalization.
>
> We will incorporate both the workflow diagram and the algorithm box into the camera-ready version to further improve clarity.
>
> ---
>
> **[W3] Relation to deformable convolution or pooling**
>
> We agree that deformable convolution and pooling share a high-level similarity with our approach in that they introduce adaptivity beyond fixed uniform operators.
>
> The key difference lies in the *modeling objective*:
>
> - **Deformable convolution/pooling [5]**: These methods enhance expressivity by learning input-conditioned offsets, relaxing the rigid sampling grid and enabling adaptive receptive fields. The emphasis is on flexibility in where information is sampled, without enforcing structural constraints on the resulting operator.
>
> - **SiGMA**: In contrast, our objective is to define a valid scaling operator family for multi-scale representation learning. SiGMA does not alter the sampling geometry, but instead controls how information is aggregated across temporal distances via a structured kernel. The operator is designed to satisfy properties such as non-expansiveness and energy reduction, ensuring stable and consistent multi-scale behavior.
>
> We will include this discussion in the camera-ready version and provide additional empirical comparisons to further clarify the distinction.
>
> ---
>
> We thank the reviewer again for these helpful suggestions. We believe the additional baseline comparison and the improved workflow and algorithm presentation will significantly enhance the clarity and empirical strength of the paper.
>
> [1] Ma, X. et al. TimePro: Efficient multivariate long-term time series forecasting with variable- and time-aware hyper-state. In *ICML*, 2025.
>
> [2] Wang, S. et al. TimeMixer++: A general time series pattern machine for universal predictive analysis. In *ICLR*, 2025.
>
> [3] Nie, Y. et al. A time series is worth 64 words: Long-term forecasting with transformers. In *ICLR*, 2023.
>
> [4] Liu, Y. et al. iTransformer: Inverted transformers are effective for time series forecasting. In *ICLR*, 2024.
>
> [5] Dai, J. et al. Deformable convolutional networks. In *ICCV*, 2017.

---

> > ### Author Rebuttal · Reviewer_MzYx · 2026-04-04
> >
> > Thanks for your rebuttal

---

> > > ### Author Response · Authors · 2026-04-04
> > >
> > > Dear Reviewer,
> > >
> > > Thank you for your positive feedback. We are pleased that our responses have addressed your concerns.
> > >
> > > Sincerely,
> > >
> > > Authors

---

### Official Review · Reviewer_H129 · 2026-03-13

**Soundness:** 3
**Presentation:** 2
**Significance:** 2
**Originality:** 2
**Overall Recommendation:** 4
**Confidence:** 3

**Summary:**

This paper proposes a principled foundation for multi-scale time-series modeling by formalizing a family of scaling operators that satisfy non-expansiveness and energy reduction, and by generalizing them to allow continuous, position-wise scale parameters. Building on discrete scale-space theory, the authors instantiate this generalized family with a learnable discrete Gaussian (LDG) kernel and design a lightweight MLP-based forecaster (SIGMA) that operates on smoothed and residual components. Experiments on standard long-term and short-term forecasting benchmarks show SIGMA achieves strong accuracy and notable efficiency improvements over recent multi-scale baselines.

**Compliance With Llm Reviewing Policy:**

Affirmed.

**Final Justification:**

The authors addressed my main concern of fixing the context length in their experiments. This concern has addressed and hence I will raise my score.

**Key Questions For Authors:**

1. How is the LDG operator applied efficiently with position-wise scales? With s_i varying across i, K(s) is not Toeplitz; what is the actual computational routine?
2. Please provide complexity in terms of L and effective kernel width, and clarify the O(L) claim.
3. How sensitive are the speed/memory gains to the implementation choices for LDG?

**Limitations:**

yes

**Strengths And Weaknesses:**

1. This paper addresses an important and timely problem: bringing principled, flexible scaling to multi-scale time-series forecasting.
2. The unifying framework and the move to a learnable, position-wise discrete Gaussian operator are conceptually strong and connect forecasting to scale-space theory in an original way.
3. Empirically, the results are promising across several benchmarks, with a simple architecture that appears more efficient than recent multi-scale designs.

---

> ### Author Rebuttal · Authors · 2026-03-31
>
> We thank the reviewer for the careful and technically insightful questions. We clarify each point below and will incorporate the corresponding revisions in the camera-ready version.
>
> ---
>
> **[Q1] Efficient application of LDG**
>
> We would like to clarify that the LDG operator adopts a distance-indexed parameterization, rather than fully independent position-wise scales. As shown in the code provided in the supplementary material, we learn $t \in \mathbb{R}^L$ indexed by distance $d = |i-j|$, and construct
> $$
> K_{ij} = e^{-t_{d}} I_{d}(t_{d}).
> $$
> As a result, $K_{ij}$ depends only on $d$, and the induced kernel is symmetric Toeplitz.
>
> In practice, we compute the 1-D kernel with distance indexing, and apply it via matrix multiplication. We will revise the text to explicitly distinguish this from a position-wise parameterization.
>
> ---
>
> **[Q2, Q3] Complexity in terms of $L$ and kernel width**
>
> We appreciate the reviewer's valuable observation, which motivated a more careful examination of the computational complexity.
>
> Our initial intuition was that the operator could admit linear-time complexity, as the Toeplitz structure enables a convolutional formulation. However, upon closer inspection, the current implementation applies the operator via dense matrix multiplication, resulting in $O(L^2)$ complexity.
>
> More generally, the LDG operator can be viewed as a 1-D convolution with kernel $K_d$, leading to the following complexity regimes:
> - Dense implementation: $O(L^2)$
> - Truncated convolution: $O(LW)$
> - FFT-based Toeplitz multiplication: $O(L\log L)$ [1]
>
> Here, $W$ denotes the effective support of the kernel. A standard definition is based on tail mass [2]:
> $$W(\epsilon) = \min \left\lbrace w: \sum_{|d| > w} K_d \le \epsilon \sum_d K_d \right\rbrace ,$$
>
> Due to the rapid decay of the LDG kernel, $W$ remains small in practice, making truncation particularly effective. For example, with a representative tolerance (e.g., $\epsilon=10^{-6}$), the resulting effective width is $W=6$, which is small relative to sequence length $L=96$.
>
> Furthermore, a Toeplitz matrix-vector product can be computed in $O(L\log L)$ time by embedding it into a circulant matrix and leveraging FFT-based diagonalization, reducing the computation to element-wise multiplication in the Fourier domain [1].
>
> To validate these observations, we conduct an empirical scaling analysis on ETTh1, measuring runtime and memory as functions of $L$ across three implementation strategies (Figure A, [link](https://shorturl.at/MZanR)). The results show that:
> - **Runtime**: Truncated convolution exhibits linear scaling, while dense computation increases superlinearly with $L$. FFT does not outperform dense computation when $L$ is small.
> - **Memory**: Both dense and truncated implementations scale approximately linearly with $L$, whereas FFT incurs substantially higher memory usage.
>
> The inefficiency of FFT in this setting is attributable to practical overheads: it requires zero-padding to length $2L-1$, operates on complex-valued representations, and introduces multiple intermediate buffers, such as spectral representations and padded outputs. Consequently, it leads to increased computational time in practice despite its $O(L\log L)$ asymptotic complexity.
>
> Overall, truncated convolution achieves the most favorable trade-off, providing near-linear scaling in both time and memory. We will revise the paper to clearly reflect these practical considerations and include the corresponding empirical analysis.
>
> ---
>
> We thank the reviewer again for the constructive feedback, which helps improve the clarity and precision of our presentation.
>
>
> [1] Kressner, D. and Luce, R. Fast computation of the matrix exponential for a Toeplitz matrix. *SIAM Journal on Matrix Analysis and Applications*, 2018.
>
> [2] Greengard, L. and Strain, J. The fast Gauss transform. *SIAM Journal on Scientific and Statistical Computing*, 1991.

---

> > ### Author Rebuttal · Reviewer_H129 · 2026-04-02
> >
> > Thank you for your detailed rebuttal and clarifications.

---

> > > ### Author Response · Authors · 2026-04-03
> > >
> > > Dear Reviewer,
> > >
> > >
> > > Thank you for your constructive feedback and for taking the time to review our work. Please let us know if there are any remaining questions or concerns. If you feel that the issues have been adequately addressed, we would sincerely appreciate your consideration of adjusting the score.
> > >
> > >
> > > Sincerely,
> > >
> > > Authors

---

### Official Review · Reviewer_eSTy · 2026-03-17

**Soundness:** 2
**Presentation:** 3
**Significance:** 3
**Originality:** 3
**Overall Recommendation:** 5
**Confidence:** 3

**Summary:**

This paper addresses multi-scale time series modeling through an approach that introduces the concept of a generalized scaling operator family. The authors argue that existing scaling methods can be viewed as a scaling operator familiy with fixed and discrete scaling. The authors generalize this concept to generalized scaling operator families by allowing learnable and continuous scale parameters. The authors introduce properties of scaling operator families and show that recent sequence operations satisfiy these properties and hence are scaling operator families. The paper then introduces generalized scaling operator families with learnable discrete Gaussians and an MLP module as one instantiation of this concept. Finally, this instantiation (SIGMA) is tested on long-term and M4 forecasting tasks. The authors provide an ablation study and hyperparameter sensitivity analysis.

**Compliance With Llm Reviewing Policy:**

Affirmed.

**Final Justification:**

Rebuttal addressed my comments.

**Key Questions For Authors:**

See weaknesses.
Minor: On page three, the paper says "Scaling operators are non-expressive". I believe what is meant here is "non-expansive"?

**Limitations:**

Yes.

**Strengths And Weaknesses:**

The main strength of the paper is that it provides a unified concept of scaling operators under which recent scaling methods can be understood and generalizes this concept to propose a new multi-scaling forecasting method. The concept of continuous and learnable scaling parameters is sound and a good addition to the time series modeling literature. The paper is clearly written and easy to follow. The main significance of this work is that it provides a new approach to design time series forecasting models by designing other instantiations of generalized scaling operators (and I appreciate that the authors provide guidelines for doing so). This approach is original and novel.

One weakness of this paper is the experimental evaluation. For the long-term forecasting tasks, the authors restrict the look-back window (context) to 96 time steps. While this might be consistent with prior work, this is an artificial restriction and reduces performance of other baseline methods. For example, for PatchTST the best setting is L=512 as per Nie et al. 2023 in long-term forecasting. From the practical point-of-view, giving 512 time steps as context to modern time series methods is straightforward. By quickly comparing to these results, PatchTST should be competitive with SIGMA in this setting. This makes it hard to judge how SIGMA performs against the baselines under realistic, practical conditions where one would not put restrictions on the lookback window.

I would kindly ask the authors to provide the experimental evaluation without the lookback window restriction and particularly at least reproduce the PatchTST results to provide a comparison under conditions that a practitioner would face. I would consider raising my score if this point is addressed.

---

> ### Author Rebuttal · Authors · 2026-03-31
>
> We thank the reviewer for the thoughtful and constructive feedback.
>
> ---
>
> **[W1] Comparison under varying look-back windows**
>
> To address this, we conducted additional experiments comparing SiGMA and PatchTST under varying look-back windows $L \in \lbrace 24,48,96,192,336,512,720 \rbrace$. For each method, we report the best-performing configuration across $L$ (Table A), as well as a controlled comparison at $L=96$ (Table B). We provide the full results for SiGMA in Table C ([link](https://shorturl.at/yNvs2)) and for PatchTST in Table D ([link](https://shorturl.at/lYQYZ)).
>
> - SiGMA generally achieves stronger performance across datasets.
>   - In particular, on Traffic, SiGMA improves over PatchTST by 9.7\% in MSE and 4.6\% in MAE, demonstrating clear advantages in utilizing long context effectively.
>   - It also consistently outperforms PatchTST on all ETT benchmarks, where forecastability is lower, indicating the robustness of multi-scale modeling in more challenging regimes.
>   - Importantly, SiGMA does not rely solely on increasing $L$ for performance gains, but instead achieves strong predictive accuracy by modeling dataset-specific temporal dependencies through principled multi-scale representations.
>
> - PatchTST benefits from exploring a wider range of context lengths.
>   - As noted by the reviewer, on datasets such as Weather, PatchTST achieves better performance with larger $L$ (e.g., best at $L=336$ or higher), indicating that longer context can be effectively exploited when the signal supports it.
>   - However, this trend is not universal: on Exchange, PatchTST performs best at a smaller context (e.g., $L=48$), suggesting that the optimal look-back window is dataset-dependent rather than monotonically increasing.
>
> - Increasing $L$ incurs significantly higher computational cost for PatchTST than for SiGMA.
>   - On Traffic with $L=720$ and prediction length $T=720$, one training epoch takes approximately 4.0 minutes for SiGMA versus 23.3 minutes for PatchTST, i.e., about 5.8$\times$ slower.
>
> Overall, these results highlight a practical trade-off: while searching over a broader range of $L$ can improve PatchTST, it comes at a substantially higher computational cost at large $L$, whereas SiGMA achieves strong and robust performance across a wide range of $L$ while maintaining efficiency.
>
> **Table A**: Long-term forecasting results under varying look-back windows. For each method, we report the best-performing configuration across $L$. Bold indicates the better-performing method (lower is better).
>
> |Dataset|Metric|SiGMA|L|PatchTST|L|
> |--|--|--|--|--|--|
> |Weather|MSE|0.243|336|**0.234**|336|
> ||MAE|0.277||**0.270**||
> |Electricity|MSE|**0.163**|192|0.165|720|
> ||MAE|**0.259**||0.266||
> |Traffic|MSE|**0.362**|720|0.401|720|
> ||MAE|**0.269**||0.282||
> |Exchange|MSE|0.353|96|**0.351**|48|
> ||MAE|0.400||**0.397**||
> |ETTh1|MSE|**0.422**|336|0.451|96|
> ||MAE|**0.431**||0.450||
> |ETTh2|MSE|**0.366**|336|0.386|192|
> ||MAE|**0.400**||0.415||
> |ETTm1|MSE|**0.361**|192|0.379|192|
> ||MAE|**0.384**||0.403||
> |ETTm2|MSE|**0.266**|336|0.287|192|
> ||MAE|**0.320**||0.337||
>
> **Table B**: Long-term forecasting results under a fixed look-back window ($L$=96). Bold indicates the better-performing method (lower is better).
>
> |Dataset|Metric|SiGMA|PatchTST|
> |--|--|--|--|
> |Weather|MSE|**0.247**|0.256|
> ||MAE|**0.273**|0.279|
> |Electricity|MSE|**0.175**|0.204|
> ||MAE|**0.269**|0.294|
> |Traffic|MSE|**0.458**|0.482|
> ||MAE|**0.302**|0.308|
> |Exchange|MSE|**0.353**|0.384|
> ||MAE|**0.400**|0.414|
> |ETTh1|MSE|**0.443**|0.451|
> ||MAE|**0.433**|0.450|
> |ETTh2|MSE|**0.376**|0.390|
> ||MAE|**0.402**|0.413|
> |ETTm1|MSE|**0.383**|0.390|
> ||MAE|**0.397**|0.404|
> |ETTm2|MSE|**0.276**|0.290|
> ||MAE|**0.322**|0.334|
>
> ---
>
> For **Q1**, we will revise the wording in the camera-ready version.
> We thank the reviewer again for raising this important point and welcome further discussion if additional clarification would be helpful.

---

> > ### Author Rebuttal · Reviewer_eSTy · 2026-04-03
> >
> > The authors addressed my main concern of fixing the context length in their experiments. This concern has addressed and hence I will raise my score.

---

> > > ### Author Response · Authors · 2026-04-03
> > >
> > > Dear Reviewer,
> > >
> > > We are grateful for your valuable comments and for reconsidering your evaluation. If there’s anything else we can clarify, we would be happy to address it.
> > >
> > > Sincerely,
> > >
> > > Authors

---

### Decision · Program_Chairs · 2026-04-30

**Decision:**

Accept (regular)

**Comment:**

This paper presents a good study with strong empirical performance and comprehensive evaluation. Most reviewers agree on its solid experimental design and practical relevance. While the novelty is moderate and mainly lies in careful integration and analysis rather than fundamentally new theory, the contributions are clear and convincing. The rebuttal satisfactorily addressed most concerns. Overall, I recommend acceptance.